



Ocean Science

# Better Baltic Sea wave forecasts: improving resolution or introducing ensembles?

**Torben Schmith, Jacob Woge Nielsen, Till Andreas Soya Rasmussen, and Henrik Feddersen**

Danish Meteorological Institute, Copenhagen, Denmark

**Correspondence:** Torben Schmith (ts@dmi.dk)

**Abstract.** TS1 The performance of short-range operational forecasts of significant wave height (SWH) CE1 in the Baltic Sea is evaluated. Forecasts produced by a base configuration are intercompared with forecasts from two improved configurations: one with improved horizontal and spectral resolution and one with ensembles representing uncertainties in the physics of the forcing wind field and the initial conditions of this field. Both of the improved forecast classes represent an almost equal increase in computational costs. Therefore, the intercomparison addresses the question of whether more computer resources would be more favorably spent on enhancing the spatial and spectral resolution or, alternatively, on introducing ensembles. The intercomparison is based on comparisons with hourly observations of significant wave height from seven observation sites in the Baltic Sea during the 3-year period from 2015 to 2017. We conclude that for most wave measurement sites, the introduction of ensembles enhances the overall performance of the forecasts, whereas increasing the horizontal and spectral resolution does not. These sites represent offshore conditions, in that they are well exposed from all directions, are a large distance from the nearest coast and in deep water CE2. Therefore, there is the a priori expectation that a detailed shoreline and bathymetry will not have any impact. Only at one site do we find that increasing the horizontal and spectral resolution significantly improves the forecasts. This site is situated in nearshore conditions, close to land and a nearby island, and is therefore shielded from many directions. Consequently, this study concludes that to improve wave forecasts in offshore areas, ensembles should be introduced. For near shore areas, in comparison, the study suggests that additional computational resources should be used to increase the resolution.

## 1 Introduction

Severe wave conditions affect ship navigation, offshore activities and risk management in coastal areas. Therefore, reliable forecasts of wave conditions are important for ship routing and planning purposes when constructing, maintaining and operating offshore facilities, such as wind farms and oil installations.

Waves are generated by energy transfer from surface winds that act on the sea. The energy transfer is determined by the fetch (the distance, over which the wind acts), and by the duration of the wind. For deep water waves, defined as having a wave height that is much smaller than the water depth, dissipation of the wave energy mainly occurs through internal processes, e.g., whitecapping. For shallow water waves, defined as having a wave height that is comparable to the water depth, dissipation via bottom friction and wave breaking over a shallow and sloping sea bed becomes important. Shallow water waves may also be refracted over a varying bathymetry, meaning that a correct and detailed description of the bathymetry is important for correctly forecasting waves in coastal areas and other shallow sea areas. Other factors with a potential effect on the development of waves include nonlinear wave–wave interactions, ocean currents, time-varying water depth due to variations in sea level and sea ice coverage.

The Baltic Sea is connected to the world ocean through the Danish waters via shallow and narrow straits (see Fig. 1), which allow virtually no external wave energy to be propagated into the area. The Baltic Sea consists of a number of basins with depths exceeding 100 m, separated by sills and marine areas with more moderate water depths. Furthermore, an archipelago with complicated bathymetry on very small spatial scales lies between Finland and Sweden. West-

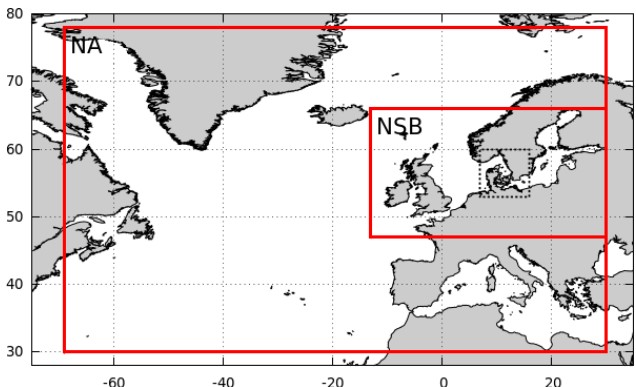

**Figure 1.** Nesting of domains in DMI-WAM. The outer frame is the North Atlantic (NA) domain, and the inner frame is the North Sea/Baltic Sea (NSB) domain. The dotted frame is the transition area. Only data from the NSB domain are analyzed in this study.

erly winds generally predominate over the area, and the most prominent causes of severe wind and wave conditions are low pressure systems that pass eastward over central Scandinavia. Winter ice occurs in the northern and eastern parts of the Baltic Sea. There is no noticeable tidal amplitude, nor are there any permanent current systems.

Short-term forecasting of surface waves is undertaken using a wave model, which is forced with forecasted wind from an atmospheric numerical weather prediction (NWP) model. The equations of the NWP model are discretized on a horizontal grid with a certain spatial resolution, which influences the maximum spatial resolution of the wave model. The available computer resources limit the horizontal grid spacing that can be afforded.

Over time, technical development has increased the available computational resources, which have traditionally been used to increase the horizontal spatial resolution of the NWP and wave models. This allows for an improved description and forecasting of the synoptic and mesoscale atmospheric systems, including the details of the associated wind field. In addition, a more detailed description of the bathymetry improves the correct description of the dissipation and refraction of waves, as argued above. Additional computer resources may also be used to improve the spectral resolution in the wave model; this includes the directional resolution and the number of frequencies included.

Increasing computer resources have also made ensemble NWP possible. The purpose of ensemble forecasts is to improve forecast skill by taking both the initial error of the forecast and the uncertainty of the model physics into account. Furthermore, ensemble forecasts allow for probabilistic forecasts, which are identified as a priority for operational oceanography (She et al., 2016), and allow for the quantification of forecast uncertainty. Ensemble wave forecast systems have been implemented at a global scale (Alves et al., 2013; Cao et al., 2009; Saetra and Bidlot, 2002) and more region-

ally in the Norwegian Sea (Carrasco and Saetra, 2008), as well as in the German Bight and western Baltic (Behrens, 2015).

From the discussion above, it is evident that additional computer resources can be used in different ways to change the wave forecast setup in order to increase the forecast quality. The purpose of the present study is to investigate the effect on the forecast quality of increasing the horizontal resolution and the spectral resolution vs. introducing ensemble forecasts. This is undertaken by verifying the DMI (Danish Meteorological Institute) operational forecasting of wave conditions in the Baltic Sea in different configurations against available observations of significant wave height.

Increasing the horizontal resolution of the NWP-system may also lead to improved wind forecasts, due in particular to better descriptions of processes in extratropical cyclones. In these cases, where the wind field is strong and varies on a small spatial scale, wave forecasts may also be improved by running the wave model in a similarly high resolution.

This paper is arranged as follows. Section 2 describes the model and setup, Sect. 3 describes the observations used and the verification methodology is described in Sect. 4. The verification of DMI-HIRLAM (Danish Meteorological Institute–High Resolution Limited Area Model) wind forecasts is in Sect. 5, whereas the verification of the significant wave height (SWH) is presented in Sect. 6. Results of the verification are discussed in Sect. 7, and conclusions are made in Sect. 8.

## 2   Model and setup

The DMI operational wave forecasting system DMI-WAM (Danish Meteorological Institute wave model) uses the third generation spectral wave model WAM Cycle4.5.1 (Günther et al., 1992), with one minor change to the source term functions. To speed up wave growth from calm seas, the spectral energy has a lower limit corresponding to a wave height of 7 cm. It is forced by the regional NWP model DMI-HIRLAM and the global NWP model ECMWF-GLM (European Centre for Medium Range Weather Forecasting–global model). WAM solves the spectral wave equation, and calculates the wave energy as a function of position, time, wave period and direction. Derived variables, such as the SWH, are calculated as suitable integrals of the wave energy spectrum.

The DMI-WAM model system forecasts waves in a larger area than just the Baltic Sea; therefore, the model setup consists of two nested spatial domains of different geographical extents (see Fig. 1): the North Atlantic (NA) and the North Sea/Baltic Sea (NSB), from which only the forecast results from the NSB domain are analyzed in this study. The NA domain uses the JONSWAP (Joint North Sea Wave Project) wave spectrum for fully developed wind-sea (Hasselmann et al., 1973) along open model boundaries, while the NSB do-

**Table 1.** Specifications of the DMI-WAM nested setup.

| Domain | North Atlantic | North Sea/Baltic Sea |
|---|---|---|
| Longitude | 69° W–30° E | 13° W–30° E |
| Latitude | 30–78° N | 47–66° N |
| Atmospheric forcing | ECMWF-HRES | DMI-HIRLAM |
| Boundary condition | JONSWAP | One-way nested |
| Depth-induced wave breaking | No | Yes |

main uses modeled wave spectra from the NA domain at its open boundaries (one-way nesting).

The wave energy is discretized into a number of wave directions and frequencies. To facilitate wave growth from calm seas, a lower limit is applied to the spectral energy. The resulting surface roughness parameterizes the effect of capillary waves, and corresponds to a minimum significant wave height of 7 cm.

The energy source is the surface wind. The sink terms are wave energy dissipation through wave breaking (white capping), wave breaking in shallow areas and friction against the sea bed. Depth-induced wave breaking (Battjes and Janssen, 1978) is only used in the NSB domain, as in the NA domain, the depth maps are not detailed enough for the activation of this effect. The wave energy is redistributed spatially by wave propagation and depth refraction, and spectrally by nonlinear wave–wave interaction. Interaction with ocean currents and effects due to varying sea levels caused by tides or storms are not incorporated.

In addition to a land mask, we have a time-varying ice mask. Below an ice concentration of 30 % sea ice is assumed to have no effect, whereas above an ice concentration of 30 % no wave energy is generated or propagated, i.e., the effect is like that of land. The applied sea ice concentrations originate from OSISAF (Satellite Application Facility on Ocean and Sea Ice; http://osisaf.met.no/p/ice/, last access: December 2017) with a frequency of 24 h and around a 25 km true horizontal resolution, gridded to ∼ 10 km horizontal resolution and interpolated to the WAM-grid. The ice cover is initialized every day at 00:00Z, and kept constant throughout each forecast run.

The surface wind forcing is provided by different atmospheric models for the two domains. For the NA domain, wind is provided by the ECMWF-HRES (European Centre for Medium Range Weather Forecasting–high resolution forecasts) global weather forecast every 3 h. For the NSB domain, the surface wind is provided every hour by DMI-HIRLAM. Setup details are summarized in Table 1.

Each forecast run is initialized using the sea state at analysis time, calculated by the previous run as a 6-hour forecast. The operational DMI-WAM suite is run four times a day at a 48 h forecast range. This is also true for the North Atlantic domain, even when new forcing is only available twice a day. This is for practical reasons, as the North Atlantic domain is

very cheap to run. Spatial fields of forecasted SWH and other variables are output at an hourly time resolution.

Historically, three different configurations of the DMI-WAM setup have been used, and data from these for the period from 2015 to 2017 are the basis for the present verification. In the old LOW configuration, the horizontal resolution is around 50 km in the NA domain and approximately 10 km in the NSB domain. The wave energy is resolved in 24 directions and at 32 frequencies, corresponding to wave periods between 1.25 and 23.94 s and wave lengths between 2.4 and 895 m (in deep water). The bathymetry is ETOPO (Amante and Eakins, 2009) in the NA domain, and the Baltic bathymetry from IOW (https://www.io-warnemuende.de/topography-of-the-baltic-sea.html, last access: March 2011) supplemented by depth data from the Danish Geodata Agency (DGA) is used in the NSB domain. More recently, an ensemble configuration (LOWENS) has been introduced that has characteristics identical to LOW but uses a parallel run of 11 ensemble members forced with perturbed atmospheric fields (initial conditions and physics). Finally, in the recently introduced HIGH configuration, the horizontal resolution is around 25 km in the NA domain and around 5 km in the NSB domain. The wave energy is resolved in 36 directions and 35 frequencies, corresponding to wave periods between 0.94 and 23.94 s, and wave lengths between 1.37 and 895 m (in deep water). Bathymetry is RTopo (Schaffer et al., 2016).

All configurations are forced by winds from ECMWF-HRES in the NA domain and DMI-HIRLAM in the NSB domain. In the NSB domain, the LOW and HIGH configurations are forced by the S03 version (3 km horizontal resolution), while LOWENS configuration is forced by the S05 version (5 km horizontal resolution). The S03 and S05 versions of DMI-HIRLAM were used operationally by DMI as deterministic and ensemble weather forecast models in the 2015–2017 period. While the better resolution of S03 might have an impact on forecasts where orographic effects are important, the impact on wind forecasts over sea is expected to be insignificant. The DMI-HIRLAM winds are interpolated to the WAM grids by bilinear interpolation. To diminish coastal effects, DMI-HIRLAM delivers a special "water-wind" to DMI-WAM, in which the surface roughness everywhere is assumed to be that of water. This enhances the wind speed in the coastal zone, which is most important in semi-enclosed areas (bays, fjords, etc.). This is basically a method of sharpening the land–sea boundary, reducing the influence of land roughness on nearshore winds. An overview of the DMI-WAM configurations is provided in Table 2.

When replacing the LOW forecast configuration with the HIGH configuration, the required computational resources for running DMI-WAM are increased by a factor of $2^2$ (increase in horizontal resolution) $\times$ 1.75 (effective decrease in time step) = 7 due to higher spatial resolution, and by a factor of 1.5 (increase of number of directions) $\times$ 35/32 (increase of number of spectral frequencies) = 1.6. This gives a to-

**Table 2.** Details of DMI-WAM configuration used in this study.

| | DMI-WAM horizontal resolution (km) | | No. wave directions | No. wave spectral frequencies | Bathymetry | | Atmospheric horizontal resolution (km) | | Ensemble members | |
|---|---|---|---|---|---|---|---|---|---|---|
| | NA | NSB | | | NA | NSB | NA (ECMWF) | NSB (DMI-HIRLAM) | NA | NSB |
| LOW | 50 | 10 | 24 | 32 | ETOPO | IOW/DGA | 16 | 3 | – | – |
| LOWENS | 50 | 10 | 24 | 32 | ETOPO | IOW/DGA | 16 | 5 | – | 11 |
| HIGH | 25 | 5 | 36 | 35 | RTopo | RTopo | 16 | 3 | – | – |

tal factor of $7 \times 1.6 \approx 11.5$. From the LOW to the LOWENS configuration, it is increased by a factor of 11 (number of ensemble members). Since these increases in computational effort are very similar, an intercomparison can contribute to answering the question of whether additional computer resources should be used to increase the spatial and spectral resolution, or to sample the uncertainty in meteorological conditions using ensembles.

The LOW and HIGH configurations both produce a class of deterministic forecast, which are also named LOW and HIGH, respectively. The LOWENS configuration produces a class of probabilistic forecast, called LOWENS. In addition, the ensemble mean defines a class of deterministic forecasts, called LOWENSMEAN.

To illustrate the differences expected among the deterministic forecasts, in Fig. 2 we show 48 h forecasts of SWH valid at the peak of the "Toini" storm on 10 January 2017. All three forecasts agree regarding the gross features of the forecasted SWH field. However, there are differences, e.g., northeast of the island of Gotland, the area with a SWH above 6 m extends further southward in the LOWNSMEAN forecast than in the LOW and HIGH forecasts.

## 3 Observations

Observed series of SWHs from wave measurement sites in the Baltic Sea, obtained from the Copernicus Marine Environmental Monitoring System (CMEMS) database, are used. None of the series have a continuous record over the 3-year period from 2015 to 2017. Data gaps may be due to malfunctions, maintenance or withdrawal of the instrument. The latter occurs during winter due to the possibility of ice. We selected sites with valid observations that covered more than 40 % of the study period, and those that were distributed reasonably throughout the study period. To avoid biases in the verification measures due to under- or over-representation of particular seasons, we also aimed to have approximately even coverage throughout the year.

Figure 3 and Table 3 show the positions and water depths of the wave measurement sites and the bathymetry of the Baltic Sea. Some sites did not make observations on the full hour; therefore, observations from these sites were ascribed

**Table 3.** Details of wave measurement sites.

| | Observation site | Long | Lat | Depth (m) | |
|---|---|---|---|---|---|
| | | | | Model | actual |
| A | Arkona WR | 13.9 | 54.9 | 46 | 45 |
| B | Bothnian Sea | 20.2 | 61.8 | 118 | $\sim 120$ |
| D | Darss Sill WR | 12.7 | 54.7 | 20 | 21 |
| F | Finngrundet WR | 18.6 | 60.9 | 56 | 67 |
| K | Knolls Grund | 17.6 | 57.5 | 63 | 90 |
| N | Northern Baltic | 21.0 | 59.2 | 68 | $\sim 100$ |
| V | Vahemadal | 24.7 | 59.5 | 18 | 5 |

to the nearest full hour, if the time distance between the observation time and the full hour was less than 15 min, otherwise they were not used. All observation series used are shown in Fig. S1 in the Supplement. The frequency of the observed SWH in different intervals for each site is given in Table 4.

## 4 Verification methodology

In this section, a short overview of the verification procedure is given. For background and more details regarding the verification measures, we refer the reader to Jolliffe and Stephenson (2003).

For each measurement series of SWHs, the corresponding forecast series for all forecast classes and for the forecast range 0 to 48 h for the grid point nearest to the position of the wave measurement site was extracted from the model output.

For the deterministic and continuous forecast classes (LOW, LOWENSMEAN and HIGH), we use the conventional performance measures root mean square error (RMSE), defined as the square root of the time average of the sum of squared differences between forecast and observation, the bias (BIAS) and the correlation coefficient (CC):

$$\mathrm{RMSE}(\tau) = \langle (h_{s,\mathrm{fcst}}^{\tau} - h_{s,\mathrm{obs}})^2 \rangle; \tag{1}$$

$$\mathrm{BIAS}(\tau) = \langle h_{s,\mathrm{fcst}}^{\tau} - h_{s,\mathrm{obs}} \rangle; \tag{2}$$

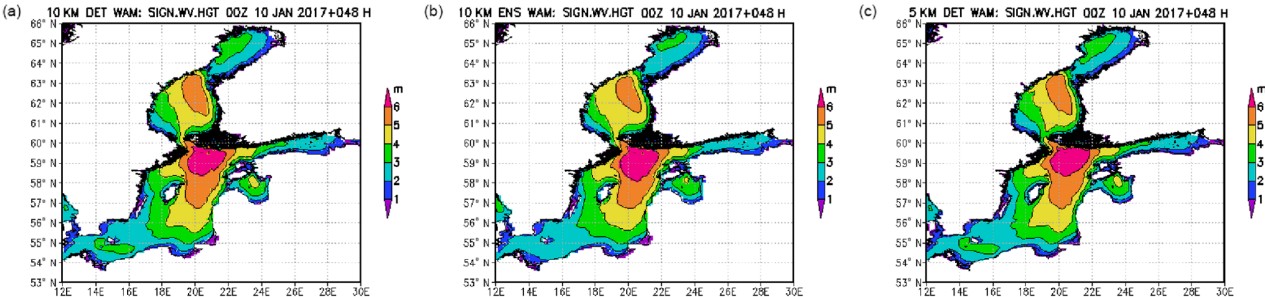

**Figure 2.** Forecasted (48 h) SWH at the peak of the "Toini" storm 10 January 2017 00:00Z for the LOW **(a)**, LOWENSMEAN **(b)** and HIGH **(c)** forecasts.

**Table 4.** Observed frequency of SWH in different bins for wave measurement sites.

| SWH (m) | 0–1 | 1–2 | 2–3 | 3–4 | 4–5 | >5 |
|---|---|---|---|---|---|---|
| Arkona WR | 0.47 | 0.39 | 0.12 | 0.01 | <0.01 | <0.01 |
| Bothnian Sea | 0.46 | 0.38 | 0.12 | 0.02 | 0.01 | <0.01 |
| Darss Sill WR | 0.67 | 0.31 | 0.02 | <0.01 | <0.01 | <0.01 |
| Finngrundet WR | 0.69 | 0.27 | 0.04 | 0.01 | <0.01 | <0.01 |
| Knolls Grund | 0.62 | 0.31 | 0.06 | 0.01 | <0.01 | <0.01 |
| Northern Baltic | 0.39 | 0.37 | 0.18 | 0.05 | 0.01 | <0.01 |
| Vahemadal | 0.78 | 0.20 | 0.02 | <0.01 | <0.01 | <0.01 |

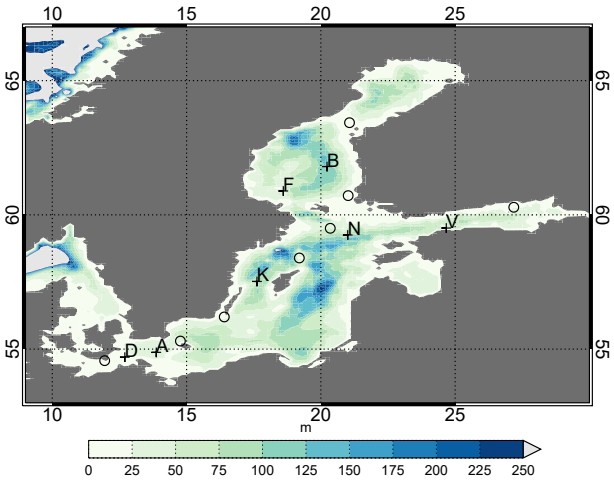

**Figure 3.** Map of the Baltic Sea with bathymetry and the positions of wave measurement sites marked with crosses. For details about the sites see Table 3. Meteorological stations used in the wind verification of DMI-HIRLAM are marked with circles.

$$\mathrm{CC}_{\text{TS2}} = \frac{\langle (h_{s,\text{fcst}}^\tau - \langle h_{s,\text{fcst}}^\tau \rangle)(h_{s,\text{obs}} - \langle h_{s,\text{obs}} \rangle) \rangle}{\sqrt{\langle (h_{s,\text{fcst}}^\tau - \langle h_{s,\text{fcst}}^\tau \rangle)^2 \rangle \langle (h_{s,\text{obs}} - \langle h_{s,\text{obs}} \rangle)^2 \rangle}}, \quad (3)$$

where $h_{s,\text{obs}}$ is the observed SWH and $h_{s,\text{fcst}}^\tau$ is a corresponding forecast with forecast range $\tau$.

The RMSE is a positive definite quantitative measure, and smaller values mean a better forecast. The bias can take positive and negative values, and a good forecast has a numerically small value. The averaging, indicated by $\langle \cdot \rangle$, is found based on all available values during the 3-year period. Also, the RMSE and BIAS as a function of $h_{s,\text{obs}}$ will be considered.

A framework for verifying probabilistic forecasts is the continuous ranked probability score CE3 (CRPS), defined as follows:

$$\mathrm{CRPS}(\tau) = \left\langle \int [F^\tau(h_s) - H(h_s - h_{s,\text{obs}})]^2 \mathrm{d}h_s \right\rangle, \quad (4)$$

where $F^\tau(h_s)$ is the forecasted probability distribution, $h_{s,\text{obs}}$ is the observed value and $H(\cdot)$ is the Heaviside step function. A small CRPS occurs when the median values of the probabilistic forecasts are close to the observed values. Also a sharp probabilistic forecast with a small spread favors a small CRPS. This means that the best forecast is achieved when CRPS is small. CRPS can be applied to both the probabilistic forecast class LOWENS, as well as the deterministic forecast classes, LOW, LOWENSMEAN and HIGH, as these can be regarded as probabilistic forecasts with a step probability distribution. For the deterministic forecast classes, the CPRS equals the mean absolute error.

Besides the continuous and probabilistic forecasts, the binary forecast of the SWH exceeding a specified threshold is also considered. The performance measure used is the Brier score, defined as follows:

Please note the remarks at the end of the manuscript.

$$BS(\tau) = \langle (p-x)^2 \rangle, \tag{5}$$

where $p$ is the forecasted probability with the forecast range $\tau$ of exceeding the threshold and $x$ takes the value of one or zero dependent on whether the threshold actually was ex-
5 ceeded or not. Thus, the Brier score is a positively definite measure, where values are between zero and one, and the lower the value, the better the forecast.

**Calculation of confidence bands** `TS3`

All the measures described above are subject to sampling un-
10 certainty; if they had been calculated using data from a time period other than 2015–2017, they would have had different values. To estimate this sampling uncertainty and thereby obtain confidence bands, we applied a block bootstrapping procedure, where a large number of resampled series with
15 the same length as the original series (3 years) were created. A blocking length of 1 month was chosen. This choice takes the atmospheric decorrelation timescale of a few weeks into account and it allows a large number of different resampled series to be made.
Each resampled series is constructed as follows: the resampled series contains three January months, and each of these months is randomly chosen, with replacement, from the three January months from the original series. A similar procedure applies for February, etc. In this way, the resampled
series are most likely different but the annual cycle is preserved. Both the observed series and the forecast series are resampled. For each pair of resampled series, bootstrapped values of the performance measures are calculated. Repeating the resampling procedure, we obtain 1000 resampled val-
ues of the measures, from which their approximate statistical distribution and confidence bands can be calculated. As a standard, confidence bands (5 % and 95 %) are calculated using the bootstrap procedure described above, and this allows for a quantitative intercomparison of the performance
measures for the different forecast classes; if the confidence bands do not overlap there is a significance difference.

## 5  Verification of the wind forecasts

In order to illustrate the benefit of the meteorological ensemble on wind forecasts the S03 deterministic and S05 ensem-
40 ble mean values were verified against available wind observations for eight coastal meteorological stations around the Baltic Sea (Fig. 3). The RMSE of all stations for the 1 January 2015–31 December 2017 period is shown in Fig. 4 as a function of forecast range. This reveals that the S05 ensem-
45 ble mean is more accurate than S03, especially at the longer forecast ranges. Similar results are found for other verification scores, such as the correlation and hit rate (not shown).

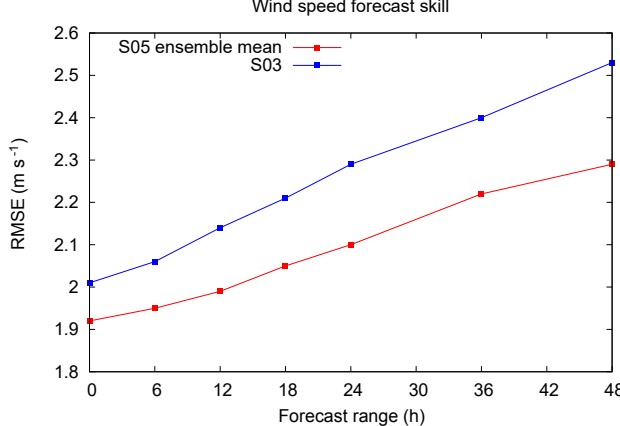

**Figure 4.** Verification of wind speed. Average RMSE between model and observations for eight coastal meteorological stations in the Baltic Sea area.

## 6  Verification of forecasted SWHs against observations

### 6.1  Deterministic measures

To get an idea of the overall quality of the forecasts, Fig. 5 50 shows scatterplots between 24 h forecasted and observed SWHs for the Bothnian Sea station. The points are distributed along the diagonal in all three configurations with correlation coefficients above 0.9. The RMSE is 0.33 m for both LOW and HIGH but is lower at 0.29 m for the 55 LOWENSMEAN forecasts, which also have the numerically lowest bias. Furthermore, for other sites, such as Arkona WR (see Fig. 6), the RMSE for the LOWENSMEAN forecasts is lower than for the LOW and HIGH forecasts; similar results are also observed for the bias. However, the scatterplot appears to be different for Arkona WR, as there is a tendency 60 to overpredict high waves in all three forecast classes.
    We now turn to the RMSE as a function of the forecast range for all sites, the plots of which can be found in Fig. S2. For all sites, the RMSE increases slightly as function of the forecast range. All sites except Vahemadal exhibit 65 qualitatively similar behavior: the RMSE for the LOW and HIGH forecasts are similar, while the RMSE is lower for the LOWENSMEAN forecasts. Thus, for the Arkona WR (shown in Fig. 7a), Bothnian Sea and Darss Sill WR sites, 70 the RMSE of the LOW and the HIGH forecasts have overlapping confidence bands. The RMSE for LOWENSMEAN gradually diverges to a lower value (around 5 cm), and for large forecast ranges the confidence bands do not overlap with those of the LOW and HIGH forecast classes. The re- 75 maining sites except Vahemadal behave similarly, but with overlapping confidence bands even for the largest forecast ranges.
    The Vahemadal site (Fig. 7b) displays different behavior. For this site, the HIGH forecast class has a significantly 80 smaller RMSE and the confidence bands do not overlap with

Please note the remarks at the end of the manuscript.

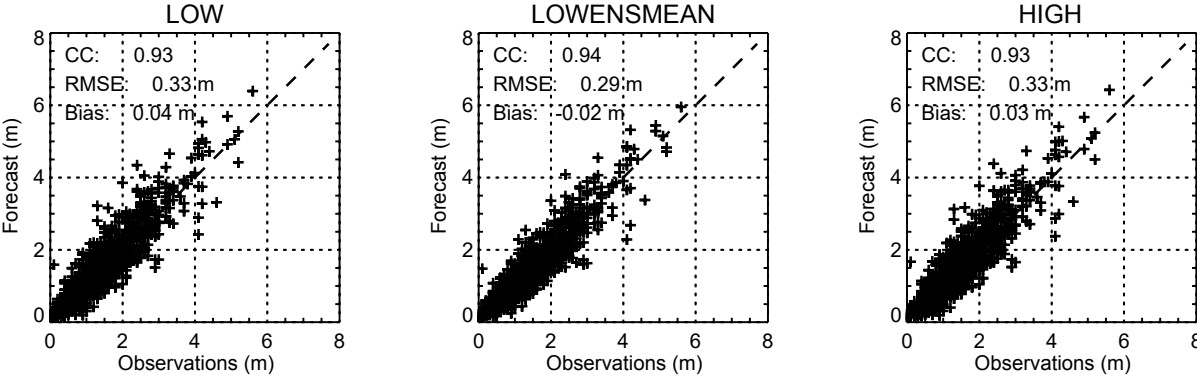

**Figure 5.** Scatterplot of the 24 h forecasts and the corresponding observations of SWH at the Bothnian Sea site for the LOW, LOWENSMEAN and HIGH forecast classes. The dotted line is the diagonal, representing a 1 : 1 agreement between the observations and the model.

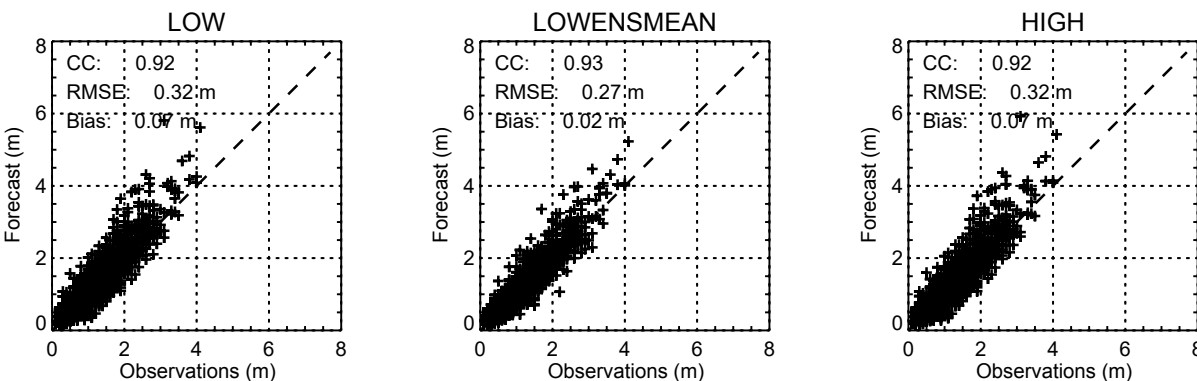

**Figure 6.** As for Fig. 5 but for the Arkona WR site.

those of the RMSE of the LOW and LOWENSMEAN forecasts. This site also has a non-negligible bias of around 12 cm for the HIGH and around 20 cm for the LOW and LOWENS-MEAN forecasts; this bias is independent of forecast range (not shown).

### 6.1.1 Performance depending on observed SWH

The RMSE of the forecasts depends on the magnitude of the SWH. Plots for all sites for the 24 and 48 h forecast ranges of RMSE as function of the SWH can be found in Figs. S3 and S4. The RMSE for Arkona WR and Vahemadal as a function of the SWH for the 48 h forecast range is shown in Fig. 8. The RMSE increases as a function of the observed SWH for both sites. For Arkona WR, the LOWENSMEAN forecast class has the lowest RMSE, although the confidence bands overlap with those from the other forecast classes. This behavior is seen at all sites, except Vahemadal. For Vahemadal, the HIGH forecast class has the lowest RMSE, and a SWH of up to 2 m; the confidence band for the HIGH class is also well separated from the confidence bands of the other forecast classes.

The bias also depends on the SWH. Plots for all sites for the 24 and 48 h forecast ranges of the bias as function of the SWH are displayed in Figs. S5 and S6. For a small SWH, the bias is close to zero for most sites. For some sites, the bias remains close to zero for increasing SWHs, as shown for Arkona WR in Fig. 9a, while for others it becomes different from zero for large SWH values. There is no noticeable difference in the bias of the different forecast classes, except for Vahemadal, shown in Fig. 9b, where the HIGH forecast class has a significantly smaller underprediction bias than the other forecast classes.

### 6.1.2 Forecasts during the "Toini" storm

The Toini storm on 11 January 2017, during which a SWH of 8.0 m was recorded at the Northern Baltic station (Björkqvist et al., 2017a), lies within our verification period. Figure 10 shows the observed SWHs at the Northern Baltic station during the 10–13 January 2017 period, i.e., including the Toini storm, which peaked in the early hours of 12 January, in addition to the corresponding 48 h forecasts. In this case there is no apparent "best" forecast. Near the peak of the storm, LOWENSMEAN performs best, but both before and after

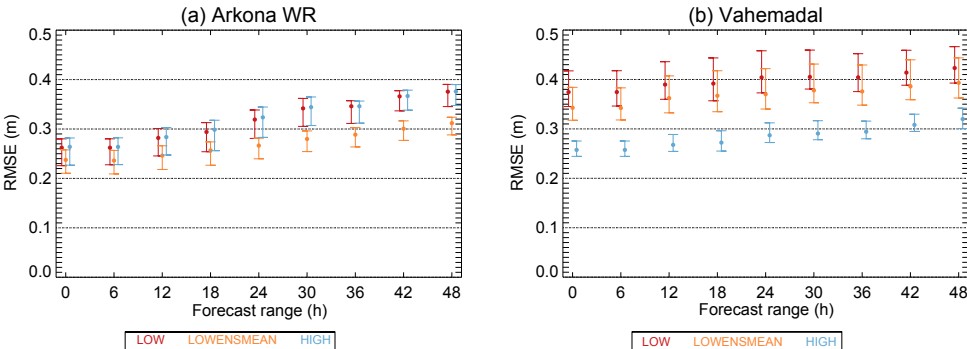

**Figure 7.** RMSE for selected forecast ranges for the Arkona WR **(a)** and Vahemadal site **(b)** for the LOW, LOWENSMEAN and HIGH forecasts. Error bars show the 5 % and 95 % confidence bands calculated by bootstrapping.

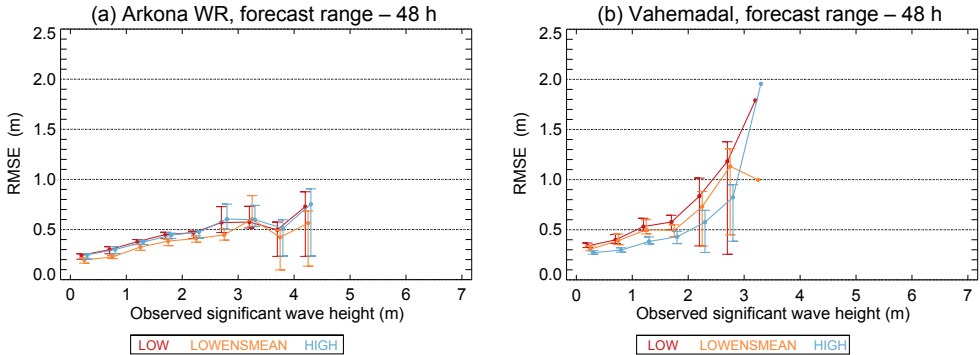

**Figure 8.** CE4 RMSE as function of SWH for the Arkona WR **(a)** and Vahemadal sites **(b)** for the LOW, LOWENSMEAN and HIGH forecasts with a forecast range of 48 h. Error bars show the 5 % and 95 % confidence bands calculated by bootstrapping.

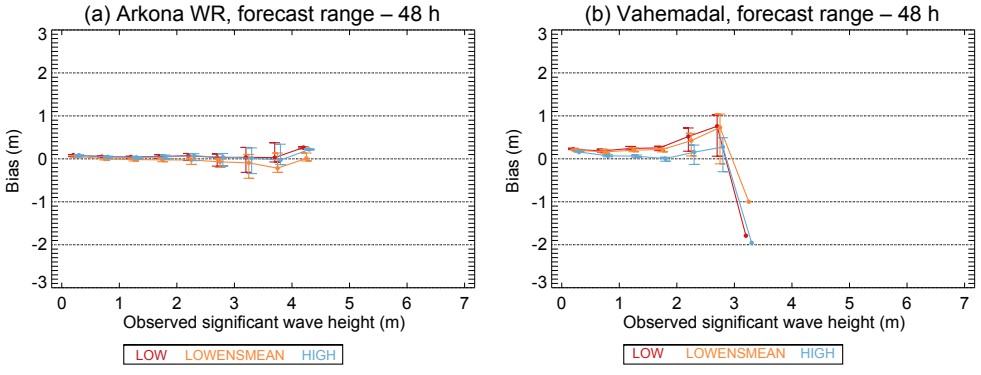

**Figure 9.** Bias as function of SWH for the Arkona WR **(a)** and Vahemadal sites **(b)** for the LOW, LOWENSMEAN and HIGH forecasts with a forecast range of 48 h. Error bars show the 5 % and 95 % confidence bands calculated by bootstrapping.

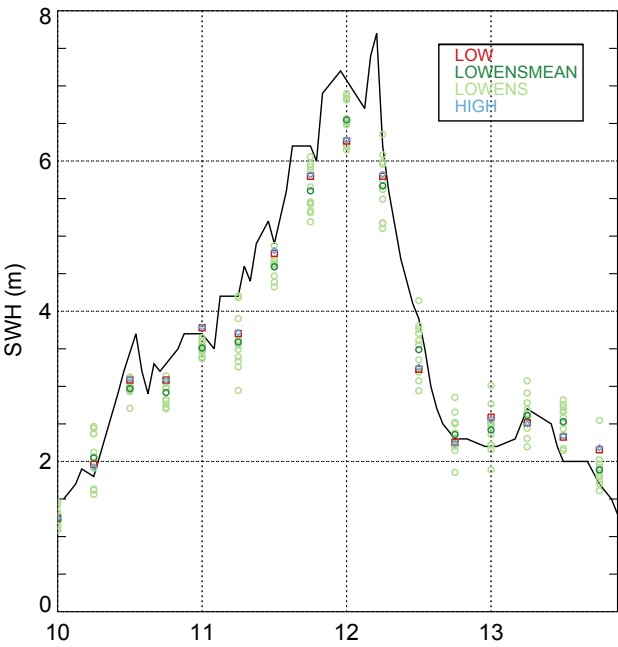

**Figure 10.** Observed SWH for the northern Baltic during the period from 10 to 13 January 2017, which included the Toini storm. Open circles are 48 h forecasts.

the storm, the HIGH/LOW forecast classes perform better. Furthermore, the LOW and HIGH forecasts are very similar in most cases, indicating that a higher resolution does not improve the forecasts. Finally, we note that the observations are generally within or just a little outside of the range of the ensemble forecast.

### 6.2   Probabilistic metrics

The 11 ensemble members of the LOWENS forecast class define a statistical distribution function, which is a probabilistic forecast of the wave conditions. The LOW, LOWENSMEAN and HIGH deterministic forecast classes may be regarded as probabilistic forecasts with a probability of one for the deterministically forecasted future state and a probability of zero for all other states.

As described in Sect. 4, we use CRPS to describe the performance of probabilistic forecasts. CRPS for selected forecast ranges can be found in Fig. S7 for all sites. As typical examples, Fig. 11 displays plots for Arkona WR and Vahemadal. All sites except Vahemadal behave qualitatively similar to the Arkona WR site: the LOWENSMEAN forecast class has a lower CRPS compared to both the HIGH and LOW classes, although the difference is only significant (non-overlapping confidence bands) for the Arkona WR, Bothnian Sea and Darss Sill WR sites, and only for the largest forecast ranges. Furthermore, for all of these sites, the LOWENS forecast class has an even lower CRPS, with confidence bands that are separated from those of the other fore-

casts classes. Again, Vahemadal behaves differently; here the HIGH forecast class has the best performance in terms of CRPS. However, for large forecast ranges, the LOWENS forecast class tends to perform equally well.

### 6.3   Binary forecasts

For the probabilistic LOWENS forecast class, a binary forecast can be derived as the probability of exceeding a defined threshold of SWH. For the deterministic forecast classes, LOW, LOWENSMEAN and HIGH, this probability of exceedance is either zero or one. As described in Sect. 4, the Brier score is used as performance measure for probabilistic, binary forecasts.

The Brier score as a function of threshold is shown for all sites in Figs. S8 and S9. Figure 12 shows the Brier score as a function of the threshold for Arkona WR and Vahemadal for the 48 h forecast range. For Arkona WR, the Brier score for the LOWENS forecast class is the smallest; however, the confidence intervals overlap with confidence intervals from the other forecasts above the 2 m threshold. Furthermore, the LOWENSMEAN forecast class has a low Brier score. This behavior is common to all sites except Vahemadal. For Vahemadal, the Brier score is smallest for the HIGH forecasts for thresholds above 1 m.

### 6.4   Rank histogram

Rank histograms serve the purpose of illustrating the reliability of probabilistic ensemble forecasts. They are a histogram of the rank of the observation, when the observation and all ensemble members of the corresponding forecast are pooled together. If the observations and the ensemble members belong to the same distribution, then the rank histogram is flat, while a U-shaped histogram indicates an unrealistically small variance within the ensemble members. For more information we refer the reader to Jolliffe and Stephenson (2003).

Rank histograms for all wave measurement sites for the 24 and 48 h forecast ranges are shown in Figs. S10 and S11, respectively. We note that all histograms show the U-shape, indicating an unrealistically small variance within the ensembles. For most sites the U-shape is symmetric, except for Vahemadal, where the U-shape is strongly asymmetrical. This corresponds well with the bias mentioned in Sect. 6.1.

### 7   Discussion

Our main finding in the previous section is that for most wave measurement sites included in this study, the LOWENSMEAN CE5 and the LOWENS forecast classes in many cases have a better performance than the LOW and HIGH forecast classes. Only one site displays differing results: the HIGH forecast class shows superior performance. The conclusions hold, whether based on overall RMSE, CRPS or the Brier score.

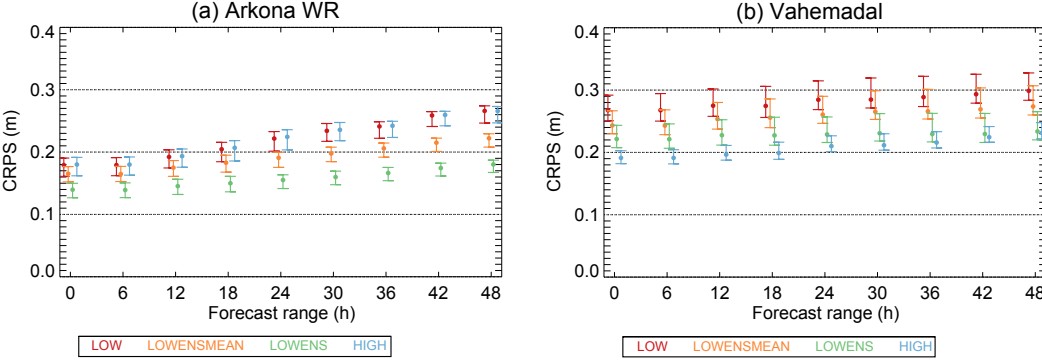

**Figure 11.** CRPS for selected forecast ranges for the Arkona WR **(a)** and Vahemadal sites **(b)** for the LOW, LOWENSMEAN, LOWENS and HIGH forecasts. Error bars show the 5 % and 95 % confidence bands calculated by bootstrapping.

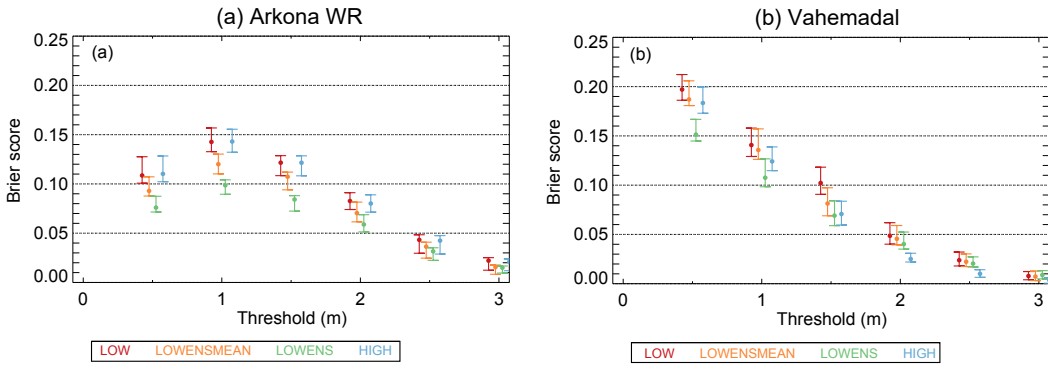

**Figure 12.** Brier score for the Arkona WR **(a)** and Vahemadal sites **(b)** for binary forecasts for a forecast range of 48 h.

In the discussion below, it should be mentioned that improving wave forecasts is not the only driving factor in reducing the grid size of the wave model. Coupling the wave model with atmosphere or ocean circulation models may give a better description of vertical fluxes of heat and momentum (Cavaleri et al., 2012). For instance, Alari et al. (2016) documented a significant improvement of modeled sea-surface temperatures by the NEMO circulation model in the Baltic Sea when two-way coupling was introduced to the wave model WAM. Introducing such coupling may demand a high horizontal resolution, in atmosphere, wave and ocean models, in order to describe the fluxes satisfactorily. Note also that the methodology applied in this study is a site-specific verification and intercomparison of the different forecast families. This is a valid approach, as most uses of the wave forecasts are site specific. However, one must keep in mind, that this approach runs the risk of underestimating the overall performance due to "double-counting errors" in both space and time. We have made no attempt to assess the magnitude of this potential effect.

## 7.1 Comparison with other operational forecast systems

Multi-year verification results from two operational deterministic wave forecast systems that cover the region in focus have been published, and can be compared to the results from the present study. Both these systems are based on the third generation WAM; the system described in Tuomi et al. (2008) has a horizontal resolution of about 22 km, while the system described in Tuomi et al. (2017) has a 1 naut. mile horizontal resolution.

For certain sites, the RMSE of the 6 h forecasts of the SWH are available for at least one of the aforementioned forecast systems in addition to the DMI-WAM forecasts; thus, comparison of the systems is possible. All sites have a water depth of more than 46 m and consequently represent offshore conditions.

We remind the reader that the cases compared in Table 5 have different wind forcing and most likely also used a different version of WAM. Therefore, the figures cannot be directly compared and differences cannot be attributed to differences in horizontal resolution with certainty.

From Table 5 one can see that for the sites considered, the LOWENSMEAN forecast has the lowest RMSE. This sup-

**Table 5.** Comparison of RMSE for SWH of 6 h forecast runs for selected sites. FIMR values are from Tuomi et al. (2008) and FMI values are from Tuomi et al. (2017).

| | FIMR | FMI | DMI LOW | DMI LOWENSMEAN | DMI HIGH |
|---|---|---|---|---|---|
| Horizontal resolution WAM | ∼ 22 km | 1 naut. mile | 10 km | 10 km | 5 km |
| Horizontal resolution NWP | ∼ 22 km | 2.5 km | 3 km | 5 km | 3 km |
| Arkona WR | – | 0.28 | 0.26 | 0.24 | 0.26 |
| Bothnian Sea | – | 0.28 | 0.25 | 0.23 | 0.25 |
| Finngrundet WR | – | 0.27 | 0.24 | 0.22 | 0.23 |
| Helsinki Buoy | 0.25 | 0.26 | – | – | – |
| Northern Baltic | 0.31 | 0.26 | 0.24 | 0.23 | 0.24 |

ports the finding of this study that for offshore conditions there is no reason to improve the resolution further than that of the LOW configuration. In addition, the results emphasize the value of describing the uncertainties in the atmospheric forcing by introducing ensembles, as this leads to a lower forecast RMSE. This is also in line with our findings in the previous section.

Test runs of a few months duration of deterministic and ensemble wave forecasts of SWH for the Baltic Sea (Behrens, 2015) also show a slight improvement of the ensemble mean forecasts, compared to deterministic forecasts, and thus support our findings.

### 7.2 Limitations of the study

#### 7.2.1 Length of verification period

Operational centers typically renew their computer installations every 5–6 years with about an order of magnitude increase in performance. At DMI, a new installation was introduced in early 2016, allowing the HIGH and LOWENS configurations to replace the LOW configuration. Presently (mid-2018) the system is mid-term upgraded; this makes it appropriate to undertake the intercomparison now as a guide for any changes in the operational setup.

For this reason, the operational forecasts performed on the present system, supplemented by delayed-mode forecasts determined the 3-year verification period used in our study. A longer verification period could evidently have reduced the sampling uncertainty in the analyses and thereby sharpened the conclusions. Conversely, the 3-year verification is not short compared to the study by Bunney and Saulter (2015) or the CMEMS verification report by Tuomi et al. (2017).

#### 7.2.2 Choice of observational base

The present verification is based on observations at a near-hourly resolution from a number of sites in the Baltic Sea. Therefore, verification is not possible in the majority of the Baltic Sea, which limits the firmness of our conclusions.

SWH derived from satellite-borne altimeters (Kudryavt-seva and Soomere, 2016) offers an alternative, which could be pursued in a future study. These data have a fair spatial coverage but at the cost of a temporal resolution of 1 day or less. This means that maximum wave heights connected to severe storms may easily be missed. Nevertheless, these data have proven useful for verification in the Baltic Sea by Tuomi et al. (2011).

### 7.3 Effect of sea ice coverage

The main effect of sea ice on the formation of waves is to limit the fetch. Furthermore, when a developed wave field approaches an ice-covered area, the wind and the waves decouple, so that the waves act more like swell, propagating through ice-covered areas while losing energy by breaking up the ice cover. The WAM model does not account for such interactions, and sea ice, when dense enough, acts as a solid shield that effectively removes all local wave energy in the model. It is implicitly assumed that dense ice will also be thick enough for this to be approximately correct. In the Baltic Sea, this may not always be the case; therefore, sea ice occurrence may represent a systematic error source in the present study. Another effect of sea ice in the Baltic is that the wave observation systems are withdrawn when ice is expected. This may cause a systematic bias in the verification analysis if strong winds during winter are excluded.

Based on Copernicus sea ice charts produced by the Finnish Meteorological Institute the ice conditions for the Baltic have been evaluated. The Finnish ice charts are produced on an approximately 1 km$^2$ grid with a temporal resolution of about 1 day in the ice season. Data are available from 2010 onwards. The average ice conditions for February for all years and the 3 years in focus can be found in Fig. S12. All 3 years from 2015 to 2017, and in particular 2015, have a smaller ice cover relative to the period from 2010 to 2018.

Another way to illustrate this is by considering the Baltic Sea integrated sea ice area, depicted in Fig. 13, which shows that the years from 2015 to 2017 have the lowest sea ice area

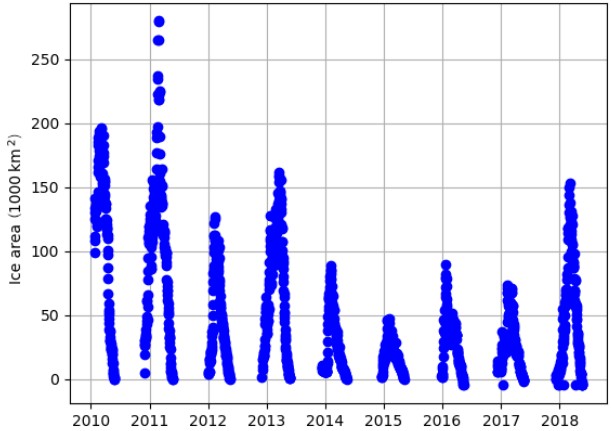

**Figure 13.** Integrated sea ice area of the Baltic Sea based on Finnish ice charts.

over the whole period from 2010 to 2018. Therefore, we may anticipate the systematic errors that arise from the occurrence of sea ice are relatively small.

## 8 Conclusions

For most sites, we find that the HIGH forecast class does not perform better than the LOW forecast class in forecasting SWH. These sites are all positioned well away from the coast in deep water and are thus freely exposed from all directions. This suggests that the resolution of the bathymetry and the spectral resolution are adequate. For these offshore sites, introducing ensembles increases the performance of the forecasts; this is seen in both the LOWENSMEAN deterministic forecasts and the LOWENS probabilistic forecasts. A similar conclusion generally holds for the binary forecast of exceeding a threshold.

For one site, Vahemedal just outside Tallin, the HIGH forecast class performs better than the other classes. The bathymetry near Vahemedal is complex and relatively shallow; thus, the bathymetry affects the wave field and an improved description of the bathymetry improves the modeled wave field. Further verification with near-coast stations may reveal whether this conclusion is general for coastal areas.

For high wave heights, there are significant systematic biases for most sites shared among all three forecast configurations. These biases are most due to model deficiencies and act to mask any differences in performance between the different forecast classes. Furthermore, the RMSE becomes large for large observed SWHs. This is expected as small timing errors in the predicted wave time series will have larger impacts on the model–observation matchup when the SWH is large. Therefore, the present study suggests that there are no indications that a further increase of the resolution of the WAM model will result in enhanced forecast performance for off-

shore conditions. In addition, the results show that introducing ensembles improves performance. This is both true for deterministic forecasts in the form of ensemble means and for probabilistic forecasts. For nearshore conditions conclusions are based on only one site, but results from this site indicate that increasing the resolution gives better forecasts, while introducing ensembles does not. This may be due to both enhanced spatial resolution, allowing for a better representation of shadow and shallow water effects, and/or spectral resolution.

Thus, the results of the present study underpin the fact that a wave model setup with an equidistant grid cannot deliver optimal wave forecasts for both coastal and offshore conditions. This is particularly true for the Baltic Sea, where very small spatial scales are found in the archipelago near the coasts of Sweden and Finland (Björkqvist et al., 2017b). Besides implementing a 0.1 naut. miles model, researchers have improved forecasts by introducing semiempirical modifications to the wave model. Cavaleri et al. (2018) also discuss this in addition to other approaches, such as one-way nesting, used in the present study (see Sect. 2), multi-cell grids (Bunney and Saulter, 2015) and triangular unstructured grids (e.g., Zijlema, 2010). These techniques may also be worth testing for the Baltic Sea.

Finally, we note the under-spread in the ensemble forecasts demonstrated in Sect. 6.4. This points to a potential for improving the combined weather–wave system.

*Data availability.* Model data are available from the authors upon request, whereas wave observations can be found on the CMEMS server (ftp://nrt.cmems-du.eu/Core/INSITU_BAL_NRT_OBSERVATIONS_013_032 TS4).

*Supplement.* The supplement related to this article is available online at: https://doi.org/10.5194/os-14-1-2018-supplement.

*Competing interests.* The authors declare that they have no conflict of interest.

*Acknowledgements.* This work was carried out under the EfficienSea2 project and supported by European Union's Horizon 2020 Research and Innovation Programme under grant agreement no. 636329. Observational data were kindly provided by the Copernicus Marine Environmental Monitoring System (CMEMS). We thank two anonymous referees for valuable suggestions and Ruth Mottram for help with the English language.

Edited by: Neil Wells
Reviewed by: two anonymous referees

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

## Remarks from the language copy-editor

## Remarks from the typesetter