# Peer review of "Better Baltic Sea wave forecasts: Improving resolution or 2 introducing ensembles?"

_Ocean Science, 2018_

## Referee Comment (RC1) · Anonymous Referee #1 · 21 Jun 2018

**1   General comments**

The paper addresses a topical issue in the operational oceanography in the marginal seas – whether to introduce ensemble forecasting. The Authors have run the wave model WAM for the Baltic Sea with different horizontal and spectral resolutions and different atmospheric forcing to study whether one should increase resolution or introduce ensembles to provide better forecast accuracy. The question is interesting, but one would expect a more thorough and systematic approach in building and introducing the model and forecast system configurations and in analysing the results.

It has been long known, although not perhaps explicitly said, that the open sea areas of the Baltic Sea, where the shallow water effects can be neglected, do not much benefit

of reducing the grid size. That said, there are several areas, where the high resolution is important to solve the shallow water effects and address the effects of islands and irregular shoreline on the wave fields.

Due to the small size of the Baltic Sea, the wave field is dominated by the wind waves and the accuracy of the wave forecast is largely dependent on the accuracy of the atmospheric forcing. Therefore comparing systems run with wind forcing from different NWP systems to address the question about choosing between ensembles and resolution is not entirely valid. Also the earlier studies the Authors refer to in the discussion most likely have different/older versions of the WAM model. Therefore the differences or non-differences cannot directly be connected to resolution or the atmospheric forcing. And also, if the time periods used in verification are relatively short (2-3 years) and different ones, the inter-annual variability in the wind conditions might also affect the accuracy. I'd expect more discussion about these subjects.

The only driving factor in reducing the grid size in the open sea wave modelling is not the accuracy of the wave forecast. There might also be other factors. For example coupling of wave and 3D ocean models might benefit of having high enough resolution. Same applies also for atmosphere – wave coupling. Furthermore, the benefits of higher resolution come also when using high-resolution wind fields nowadays available for the Baltic (e.g. HARMONIE with 2.5 km resolution), which are not possible to get full benefit from if wave model resolution is coarser. I'd also like to see more discussion related to these subjects.

Also, I think that the title should include indication, that you are focusing on the open sea, deep water areas.

Is same wind forcing used both for HIGH and LOW NSB grids? This is not explicitly said in the manuscript. And is the forcing used the deterministic ECMWF or the HIRLAM wind field? Table 1 mentions both HIRLAM and ECMWF and Table 2 only states that one ensemble member is used as forcing, but not indicated whether the 1 ensemble is

the ECMWF or HIRLAM deterministic forecast or something else. If HIRLAM is used for the HIGH and LOW NSB grids, then you are comparing wave model results with different NWP forcings against each other. Is it then question about resolution or different wind forcings? I suggest that you run both HIGH and LOW NSB grids using the control forecast from the ECMWF ENS system and compare the difference between them and the LOWENS to find what type of effects the resolution and introducing ensembles causes to the system. Furthermore, it is of course interesting to see, if with higher resolution wind forcing (e.g. HIRLAM) the results would further improve. If and when you use the HIRLAM forcing for the wave models, please specify, how you process the wind fields with 3 km resolution to the wave model grid with 5 km (and/or 10 km) resolution.

Also, you should separately check, what can be addressed to spectral resolution and what to grid resolution. And also please check other parameters than SWH, for example it would be interesting to see, if there are affects to wave periods or directions, when using higher spectral resolution.

When looking through the supplement material, I was bit confused, why Arkona and Vahemadal were chosen to be the stations shown in the manuscript. E.g. looking Fig S2 Finngrundet, Nothern Baltic, Huvudskar show that HIGH gives lower rmse in many cases for the higher (of over 3 m) significant wave heights than LOW or LOWENS. If the lower rmse of LOWENS over longer forecast ranges come mainly from forecasting smaller than 3 m SWH it might not be that useful for duty forecasters. This type of conditions typically do not affect the marine traffic or the offshore structures, it is the extremes. Therefore it would be important to see how the different forecast systems behave in high wave conditions. The time period used in this study contains at least the January 2017 storm. It would be interesting to see a detailed comparison of the results during this storm and also in some other high wind events.

It is good that the Authors have shown that with ECMWF ENS forcing the accuracy of the wave forecasts is ok in the open sea areas of the Baltic Sea. I suggest that

the authors do the more comprehensive model runs suggested above and also more detailed analysis of the results and also discuss the advantages and disadvantages of each system more thoroughly. Furthermore, it would be interesting to see how much skill the ensemble forecasts have for longer forecast period. To my experience, there is not much spread in the ensembles for the first two or three days and the true benefits of the ensemble system and probabilistic forecast usually comes with longer forecast ranges. It would be interesting to see up to which forecast lengths the ensemble system shows skill in forecasting the Baltic Sea wave conditions both in average and extreme conditions.

Please also see my specific comments given below.

**2 Specific comments**

**2.1 Introduction**

Lines 29-35: I'd expect that the concept of deep and shallow water waves is introduced here, since this is one of the key issues in the discussion of the results.

Line 33: Bathymetry is important only if waves interact with bottom.

Lines 29-25: How about weak non-linear wave-wave interactions?

Lines 41-42: Seasonal ice conditions vary quite a lot in the Baltic. Perhaps this description refers to an average ice winter?

Line 51-52: Is Baltic Sea shallow considering the average wave conditions? If then the use of higher resolution should make a difference, which is not in agreement with the conclusion drawn by the Authors later on. Baltic Sea is shallow compared to the Oceans, but when considering the typical wave periods/lengths in the Baltic, in most

cases waves in the open sea areas are deep water waves, expect for high and extreme wave conditions.

**2.2   Model and setup**

This section needs restructuring. All information needed is basically given, but the order of things and the fact that some information is only given in Tables and the table is not referred in the corresponding place in text makes it difficult to follow.

Also please define explicitly, which wind forcing is used for LOW and HIGH configurations. Table 3 indicates that deterministic ECMWF forcing is used for the coarse, larger domain and HIRLAM (and possibly also ECMWF?) for the smaller high resolution domain. It is not clear to me if this Table refers to DMI operational setup or for the setup used in this paper.

Lines 73-77: Please specify the source terms and formulations used in the model runs.

Lines 78-82: Specify the horizontal resolution of the areas already here or cite a Table where they are given. I also suggest adding the resolution info to Table 1.

Line 88: Specify the various sources used to compile the bathymetry

Line 121-122: You use only 11 members of the total 50 available from ECMWF. How do you select, which members you use?

Table 2: It is unclear to me what the column 'Ensemble members' mean for LOW ad HIGH.

**2.3   Observations**

Why not use Helsinki wave buoy data from Gulf of Finland? This should be available through CMEMS. Helsinki site mostly represent deep water conditions and it would be

interesting to see, how the setups behave there compared to Vahemadal.

Table 3 gives only model depth at the buoy locations. It would be important to know also the actual depth at the buoy locations to evaluate, whether the model is adequately able to account for the deep and shallow water features in the wave field.

It is bit unclear to me, what is the function of Figure 3. The details are lost here, since the images are so small. If they area meant to represent the overall description of the wave conditions at each site, please also (or maybe instead) give some description in the text. And if it is to show the gaps in the measured data, that could be put in a table.

**2.4   Verification**

Give some explanation, why you have selected Bothnian Sea, Arkona and Vahemadal stations for more detailed analysis

I also suggest doing some verification of the forcing wind fields.

In addition to verifying the general accuracy, I'd expect to see some verification of high wind/wave events. They are the most important ones to forecast accurately considering the marine traffic and offshore structures.

I would also expect more discussion of the importance of wind field accuracy on the accuracy of the wave forecast. The accuracy of wave forecast in the open sea areas might not benefit from higher resolution in the wave model grid, but what about when the wind forcing has high resolution, such as the HARMONIE forecasts run for the Baltic with 2.5 km resolution in several of the MET services. In order to account for the benefits of this, higher resolution in wave model grid might become important.

**2.5 Discussion**

You should very carefully analyse and explain, what you are actually comparing in Table 2. To my understanding you are comparing wave forecast system, which have different resolutions, wind forcing and also most likely different wave model versions. So the differences in accuracy cannot solely be attributed to resolution.

Table 4 – Why have you not calculated rms errors for the Helsinki wave buoy for LOW, HIGH and HIGHENS?

I'm not sure why the ice coverage is discussed here. You are comparing the forecasts against buoy measurements and the buoys are recovered well before there is a risk of ice in the area. Therefore handling of ice should not cause any problems in your verification results. That said, you of course have this element during the season and in the areas where you are unable to do the verification. You could also give a short description of the ice conditions in 2015-2017 so that readers would be able to evaluate, how big effect this might be.

**3 Technical corrections**

Page 1: line 34, has → have

Page 2, line 52: See → Sea

---

## Referee Comment (RC2) · Anonymous Referee #2 · 18 Jul 2018

OVERVIEW:

This paper will serve a useful purpose in documenting performance of an operational wave forecast modelling system for the Baltic and in assessing and discussing the relative benefits of increasing wave model resolution versus a probabilistic forecast system in this specific scenario - where approximately an order magnitude increase in computing power has been available. For such a study, the authors have done a good job with being concise in their use of probabilistic verification metrics and delivered a clear set of results.

However, I would recommend that publication is made subject to a number of major revisions. These are required in order to address a number of questions raised by the study, but which the authors have only dealt with very briefly or passed over:

[Figure]

1. For a wind driven wave model, the nature and quality of the forcing winds are a key consideration in the model performance. The driving wind model therefore needs to be well documented and, specifically for this paper, any differences in horizontal resolution associated with the deterministic and ensemble forecasts systems need to be provided clearly in Section 2 These were not clear to me on my read through, and I am left with the impression that the authors have compared a 10km wave model with a 5km wave model but using a similarly specified wind model for both deterministic and ensemble forecasts?

2. If this is indeed the case, then I think the wind forcing being used, wave model resolutions chosen and available observations naturally lean the study toward favouring the ensemble. This is acceptable, but needs to be acknowledged and discussed further within the paper. From a wind perspective, if no higher resolution atmosphere model that will improve representation of the land-sea boundary layer is available then the ensemble's provision of multiple answers will generally help the verification scores from that system. Whilst a costly enhancement, the change from (LOW) 10km to (HIGH) 5km resolution may not be enough to significantly enhance wave forecast performance in the coastal zone and, besides, only one observation site is available to illustrate coastal performance. This means that it is difficult for the reader to get a clear picture of what advantages the HIGH res model is expected to yield - I'd suggest that might be improved by some visualization of model fields in order that the impact of changes from LOW to HIGH over the wider region can at least be illustrated.

3. Although, in my view, the experiments favour the ensemble system, the paper still raises a valid point: which is that when using regular grid wave models and an order of magnitude computing resource to invest then the ensemble will likely provide a better return, in terms of improving forecast skill over the larger offshore part of the domain. However, in order to make this point the authors also need to be mindful of and discuss the study within the context of rather more of the open literature than they have done. For example, Cavaleri et al (2018) provide an exhaustive discussion of coastal

processes and how wind and wave models need to improve in order to properly represent these - it would be good if the authors can set out where and how the HIGH system attempts to address these aspects of coastal forecasting better than the LOW or LOWENS systems. Similarly, there is also the question of whether an unstructured or refined grid approach would enable significant improvements in coastal regions of the domain whilst keeping the model efficient offshore and enabling a best of both worlds approach (e.g. Bunney and Saulter,, 2016). So I would recommend that the authors try to address these aspects of the paper with appropriate references in both Sections 1 and 6.

SPECIFIC COMMENTS

Paragraphs at line 44 and 48. I think this discussion could be a bit more expansive? The authors have followed through the practical viewpoint where the wave model is scaled to the NWP and then resolution is increased if there is spare resource. This is a quite standard 'in practise' way of working, but as a motivating point it would be good if the authors could expand on what scales they believe are required for an idealised/pragmatic wave forecasting system that dealt with both coastal and offshore areas of the region.

Sentence at line 64. I'm not convinced that the ensemble vs resolution increase argument is generic, rather it depends on where the model is being used and how end-users will deal with the resulting products. So I think it would be better to contextualise this argument to the situation in question - a wind-wave dominated regional sea with a mixture of offshore and coastal regimes.

Sentence at line 112. I'm not convinced the information about the spin-up is that useful.

Paragraph at line 117. Around here would be an excellent place to add further detail regarding the NWP forcing.

Paragraphs at lines 247 and 255. The dependencies of RMSE/bias on SWH are to

be expected when matching up deterministic forecasts since small timing errors in the predicted wave time-series will have larger impacts on the model-observation match-up in the upper percentiles of the SWH pdf than in the lower percentiles.

Section 6.1. It would be useful to state the resolution of the NWP systems underpinning Tuomi et al.'s wave models.

Section 6. For completeness it would be worth discussing the spread-skill characteristics of the LOWENS system. At the sort of short forecast ranges discussed, these systems are usually under-spread and it would be useful to know if this is also the case here (and if not, why not?). The ability of the ensemble to properly generate spread provides the difference between running a system that provides some improvements to forecast verification vs a deterministic model through a partial sampling of forecast uncertainty, and one that genuinely samples the likely observed outcomes.

Section 6. This would be a good place to talk through the computational limits placed by using a regular grid scheme in this region and some of the other modelling options that might allow some best of both worlds solution to be achieved in future. Its fair to say that in supercomputing terms a resource increase of order 3-10 times might be the maximum expected over 1 or 2 new systems, so the problem highlighted here is important.

SPECIFIC CORRECTIONS

It is suggested that the following are checked as corrections for typos/grammar:

Lines 34-35. Other factors, which potentially have an effect on the development of wave include ocean currents, varying water depth...

Line 41. ...severe wind and wave conditions are low pressure systems passing eastward

Line 47. ...only a certain degree of horizontal grid spacing can be afforded

Line 52. ...Baltic Sea.

Line 61. ...(Carrasco and Saetra, 2008)

Figure 4: Caption. Dotted line is the diagonal, representing a 1:1 agreement between observations and model.

Line 351: ...increases the performance.

REFERENCES:

Cavaleri et al., 2018: Wave modelling in coastal and inner seas. Progress in Oceanography, https://doi.org/10.1016/j.pocean.2018.03.010

Bunney and Saulter, 2016. An ensemble forecast system for prediction of Atlantic-UK wind waves. Ocean Modelling, http://dx.doi.org/10.1016/j.ocemod2015.07.065

---

## Author Comment (AC1) · 20 Jul 2018

**Response to Anonymous Referee #1 with authors responses marked with \**

1 General comments The paper addresses a topical issue in the operational oceanography in the marginal seas – whether to introduce ensemble forecasting. The Authors have run the wave model WAM for the Baltic Sea with different horizontal and spectral resolutions and different atmospheric forcing to study whether one should increase resolution or introduce ensembles to provide better forecast accuracy. The question is interesting, but one would expect a more thorough and systematic approach in building and introducing the model and forecast system configurations and in analysing the results. It has been long known, although not perhaps explicitly said, that the open

sea areas of the Baltic Sea, where the shallow water effects can be neglected, do not much benefit of reducing the grid size. That said, there are several areas, where the high resolution is important to solve the shallow water effects and address the effects of islands and irregular shoreline on the wave fields.

\* It may be common knowledge that for most areas in the Baltic Sea wave forecasts will not benefit from (further) decrease of the grid size, '..although perhaps not explicitly said'. It is not clear to us, what the intention of this remark is; we find it a perfect motivation for our study.

Due to the small size of the Baltic Sea, the wave field is dominated by the wind waves and the accuracy of the wave forecast is largely dependent on the accuracy of the atmospheric forcing. Therefore comparing systems run with wind forcing from different NWP systems to address the question about choosing between ensembles and resolution is not entirely valid. Also the earlier studies the Authors refer to in the discussion most likely have different/older versions of the WAM model. Therefore the differences or non-differences cannot directly be connected to resolution or the atmospheric forcing. And also, if the time periods used in verification are relatively short (2-3 years) and different ones, the inter-annual variability in the wind conditions might also affect the accuracy. I'd expect more discussion about these subjects.

\* We agree that wind waves dominate the Baltic Sea and therefore comparing wave forecasts driven by wind forcing from different NWP-systems will not be entirely valid. However, due to an unfortunate error in table 1, there is a misunderstanding about the wind forcings in our study. We will explain this below and will revise the manuscript and in particular table 1 accordingly. Discussion of length of verification period will be included in the revised manuscript. Btw., our three-years verification period is not short compared to other published work. For instance, (Bunney and Saulter 2015) use eight months, (Pezzutto et al. 2016) use seven months, and (Cao et al. 2009) use 12 months.

The only driving factor in reducing the grid size in the open sea wave modelling is not the accuracy of the wave forecast. There might also be other factors. For example coupling of wave and 3D ocean models might benefit of having high enough resolution. Same applies also for atmosphere – wave coupling. Furthermore, the benefits of higher resolution come also when using high-resolution wind fields nowadays available for the Baltic (e.g. HARMONIE with 2.5 km resolution), which are not possible to get full benefit from if wave model resolution is coarser. I'd also like to see more discussion related to these subjects.

\* We will include a discussion of higher resolution required by two-way coupling to atmospheric model and ocean model.

\* We agree that increasing the horizontal resolution of the NWP-system may lead to better wind forecasts, due to better descriptions of processes in cyclones, etc. This does not necessarily dictate that the wave model should be run at the same high resolution. We will include some discussion on this.

Also, I think that the title should include indication, that you are focusing on the open sea, deep water areas.

\* We will change the title to emphasize the focus on the open sea.

Is same wind forcing used both for HIGH and LOW NSB grids? This is not explicitly said in the manuscript. And is the forcing used the deterministic ECMWF or the HIRLAM wind field? Table 1 mentions both HIRLAM and ECMWF and Table 2 only states that one ensemble member is used as forcing, but not indicated whether the 1 ensemble is the ECMWF or HIRLAM deterministic forecast or something else. If HIRLAM is used for the HIGH and LOW NSB grids, then you are comparing wave model results with different NWP forcings against each other. Is it then question about resolution or different wind forcings? I suggest that you run both HIGH and LOW NSB grids using the control forecast from the ECMWF ENS system and compare the difference between them and the LOWENS to find what type of effects the resolution and introducing en-

СЗ

sembles causes to the system. Furthermore, it is of course interesting to see, if with higher resolution wind forcing (e.g. HIRLAM) the results would further improve. If and when you use the HIRLAM forcing for the wave models, please specify, how you process the wind fields with 3 km resolution to the wave model grid with 5 km (and/or 10 km) resolution.

\* All configurations in the NSB-domain are forced with wind from the DMI-HIRLAM NWP-system. For the HIGH and LOW configurations the horizontal resolution is 3 km, while for the LOWENS it is 5 km. The ECMWF-forced North Atlantic domain is deterministic in all cases, and serves only as boundary data.

\* Unfortunately, there was an error in in table 1 in the submitted manuscript and this table will in the final manuscript be modified to: Table 1 Specifications of DMI-WAM nested setup. Domain North Atlantic North Sea/Baltic Sea Longitude 69W-30E 13W-30E Latitude 30N-78N 47N-66N Atmospheric forcing ECMWF GLM Hirlam S03/S05 Boundary condition JONSWAP Nested Bathymetry Rtopo Rtopo/IOW/GEO

\* We will include a sentence on how the 3/5 km HIRLAM wind fields are transformed to 5 and 10 km WAM grids by bilinear interpolation

Also, you should separately check, what can be addressed to spectral resolution and what to grid resolution. And also please check other parameters than SWH, for example it would be interesting to see, if there are affects to wave periods or directions, when using higher spectral resolution. Checking the effect of changing spectral resolution and grid size separately. This could certainly be interesting, but would require a lot of additional work. The scope of this work is to intercompare the performance of different operational configurations, rather than idealised studies aiming at isolation effects.

\* Verification analysis of wave period and wave direction, although not as extensive as for significant wave height, will be included in the revised manuscript. One example would be to show scatter diagrams as the one shown in Fig 1.

When looking through the supplement material, I was bit confused, why Arkona and Vahemadal were chosen to be the stations shown in the manuscript. E.g. looking Fig S2 Finngrundet, Nothern Baltic, Huvudskar show that HIGH gives lower rmse in many cases for the higher (of over 3 m) significant wave heights than LOW or LOWENS. If the lower rmse of LOWENS over longer forecast ranges come mainly from forecasting smaller than 3 m SWH it might not be that useful for duty forecasters. This type of conditions typically do not affect the marine traffic or the offshore structures, it is the extremes. Therefore it would be important to see how the different forecast systems behave in high wave conditions. The time period used in this study contains at least the January 2017 storm. It would be interesting to see a detailed comparison of the results during this storm and also in some other high wind events.

\* Vahemadal was chosen because it stands out against all other stations in the analysis (HIGH forecasts performs better for this station), and Arkona because for this station, together with Darss Sill, the ENSMEAN performed significantly better than the other forecast classes. For the four last stations, the ENSMEAN perform best, but this result is not statistically significant. We will follow the referees' suggestion and show stations representing the three different situations. Remember, however, that for SWH above 3 m there are few observations/forecasts and the statistics becomes very uncertain, as shown by the large error bars on fig. S2, and also mentioned in the text.

\* Introducing the 11 January 2017 'Toini' storm as a case is a good suggestion. We vill devote a section to this in the revised manuscript, including and discussing the plot shown in Fig 2 of SWH during January 2017, 48 hour forecast for Northern Baltic.

It is good that the Authors have shown that with ECMWF ENS forcing the accuracy of the wave forecasts is ok in the open sea areas of the Baltic Sea. I suggest that the authors do the more comprehensive model runs suggested above and also more detailed analysis of the results and also discuss the advantages and disadvantages of each system more thoroughly. Furthermore, it would be interesting to see how much skill the ensemble forecasts have for longer forecast period. To my experience, there is

not much spread in the ensembles for the first two or three days and the true benefits of the ensemble system and probabilistic forecast usually comes with longer forecast ranges. It would be interesting to see up to which forecast lengths the ensemble system shows skill in forecasting the Baltic Sea wave conditions both in average and extreme conditions.

\* As explained above, our runs are all forced with DMI-HIRLAM ensembles, and therefore they only reach 48 hour forecast time.

\* The experience that the benefits of an ensemble system shows up after three days only is contrary to our results (Fig S1), where the effect of ensembles for some stations shows up from about 36 hours.

Please also see my specific comments given below.

2 Specific comments 2.1 Introduction Lines 29-35: I'd expect that the concept of deep and shallow water waves is introduced here, since this is one of the key issues in the discussion of the results.

\* Agree, we will do that in the revised manuscript.

Line 33: Bathymetry is important only if waves interact with bottom.

\* Agree, we will make a more precise formulation

Lines 29-25: How about weak non-linear wave-wave interactions?

\* Agree, (Non-linear) wave-wave interaction should be mentioned here

Lines 41-42: Seasonal ice conditions vary quite a lot in the Baltic. Perhaps this description refers to an average ice winter?

\* Yes, this is included to remind the reader that sea ice is an issue in the Baltic Sea.

Line 51-52: Is Baltic Sea shallow considering the average wave conditions? If then the use of higher resolution should make a difference, which is not in agreement with

the conclusion drawn by the Authors later on. Baltic Sea is shallow compared to the Oceans, but when considering the typical wave periods/lengths in the Baltic, in most cases waves in the open sea areas are deep water waves, expect for high and extreme wave conditions.

\* We agree that we use 'shallow sea' in two different meanings throughout the manuscript 1) as opposed to the deep world ocean, and 2) the small water depth allows the wind waves to sense the bottom. This is confusing and we will clarify this in the revised manuscript

2.2 Model and setup This section needs restructuring. All information needed is basically given, but the order of things and the fact that some information is only given in Tables and the table is not referred in the corresponding place in text makes it difficult to follow.

\* We will restructure our section 2.2 to make it more understandable.

Also please define explicitly, which wind forcing is used for LOW and HIGH configurations. Table 3 indicates that deterministic ECMWF forcing is used for the coarse, larger domain and HIRLAM (and possibly also ECMWF?) for the smaller high resolution domain.It is not clear to me if this Table refers to DMI operational setup or for the setup used in this paper.

\* We will clarify the wind forcing in the different configurations, as explained above (table 1).

Lines 73-77: Please specify the source terms and formulations used in the model runs.

\* Source terms are described on page 93-99. We will consider extending the description a bit.

Lines 78-82: Specify the horizontal resolution of the areas already here or cite a Table where they are given. I also suggest adding the resolution info to Table 1.

\* Horizontal resolutions are given in Table 2, which we will refer to.

Line 88: Specify the various sources used to compile the bathymetry

\* We will specify these extra various sources.

Line 121-122: You use only 11 members of the total 50 available from ECMWF. How do you select, which members you use?

\* As explained above, we use DMI-HIRLAM atmospheric ensemble forcing, of which 11 ensemble members are run at routine basis at DMI. The ensembles are generated by perturbing a number of processes, e.g. cloud physics, which do not have a direct impact on the wind field. A subset of 11 ensembles was recommended by the DMI-HIRLAM ensemble developers to cover the spread in surface wind.

Table 2: It is unclear to me what the column 'Ensemble members' mean for LOW ad HIGH.

\* '1' means deterministic forecast. We will replace by '-'.

2.3 Observations Why not use Helsinki wave buoy data from Gulf of Finland? This should be available through CMEMS. Helsinki site mostly represent deep water conditions and it would be interesting to see, how the setups behave there compared to Vahemadal.

\* Helsinki wave buoy does not have many valid data in our verification period, compared to the stations used, see plot in Fig 3. Therefore it is not included.

Table 3 gives only model depth at the buoy locations. It would be important to know also the actual depth at the buoy locations to evaluate, whether the model is adequately able to account for the deep and shallow water features in the wave field.

\* We will list the actual water depths in table 3.

It is bit unclear to me, what is the function of Figure 3. The details are lost here, since

the images are so small. If they area meant to represent the overall description of the wave conditions at each site, please also (or maybe instead) give some description in the text. And if it is to show the gaps in the measured data, that could be put in a table.

\* We will put Figure 3 in the supplementary information; we think the reader should have the opportunity to see which data series were actually used. In the main text we will replace figure 3 by a table showing fraction of swh-height intervals in m to give the reader a feeling of the data: 0-1 1-2 2-3 3-4 4-5 ArkonaWR 0.47 0.39 0.12 0.01 0.00 BothnianSea 0.46 0.38 0.12 0.02 0.01 DarsserSWR 0.67 0.31 0.02 0.00 0.00 FinngrundetWR 0.69 0.27 0.04 0.01 0.00 Knollsgrund 0.62 0.31 0.06 0.01 0.00 NorthernBaltic 0.39 0.37 0.18 0.05 0.01 Vahemadal 0.78 0.20 0.02 0.00 0.00

2.4 Verification Give some explanation, why you have selected Bothnian Sea, Arkona and Vahemadal stations for more detailed analysis

\* As explained above, we will explain why we choose these stations.

I also suggest doing some verification of the forcing wind fields.

\* We will find and properly cite exsisting verification of HIRLAM. We think that doing extra verification of HIRLAM is a paper in itself and beyond the scope of this work.

In addition to verifying the general accuracy, I'd expect to see some verification of high wind/wave events. They are the most important ones to forecast accurately considering the marine traffic and offshore structures.

\* Sections 5.1.1 (RMSE and BIAS as function of SWH) and 5.3 (Brier score verification) are already verifications of high wave events. In addition we will include a qualitative intercomparison of the Jan 2017 storm, as stated above.

I would also expect more discussion of the importance of wind field accuracy on the accuracy of the wave forecast. The accuracy of wave forecast in the open sea areas might not benefit from higher resolution in the wave model grid, but what about when the wind forcing has high resolution, such as the HARMONIE forecasts run for the

Baltic with 2.5 km resolution in several of the MET services. In order to account for the benefits of this, higher resolution in wave model grid might become important.

\* We agree that increasing the horizontal resolution of the NWP-system may lead to better wind forecasts, due to better descriptions of processes in cyclones, etc. We will include discussion on this in the revised manuscript.

2.5 Discussion You should very carefully analyse and explain, what you are actually comparing in Table 2. To my understanding you are comparing wave forecast system, which have different resolutions, wind forcing and also most likely different wave model versions. So the differences in accuracy cannot solely be attributed to resolution.

\* We do not understand this remark on table 2.

Table 4 – Why have you not calculated rms errors for the Helsinki wave buoy for LOW, HIGH and HIGHENS?

\* Due to lack of observations, see above.

I'm not sure why the ice coverage is discussed here. You are comparing the forecasts against buoy measurements and the buoys are recovered well before there is a risk of ice in the area. Therefore handling of ice should not cause any problems in your verification results. That said, you of course have this element during the season and in the areas where you are unable to do the verification. You could also give a short description of the ice conditions in 2015-2017 so that readers would be able to evaluate, how big effect this might be.

\* The effect of ice cover is described because it is a potential systematic error source of the wave forecasts, i.e. a potential for future improvement of wave forecast in the Baltic Sea.

Fig. 1. Scatter diagram of 48 hour forecast of peak period for Northern Baltic

---

## Author Comment (AC2) · 23 Jul 2018

Response to Anonymous Referee #2 with authors' replies marked by ***

OVERVIEW: This paper will serve a useful purpose in documenting performance of an operational wave forecast modelling system for the Baltic and in assessing and discussing the relative benefits of increasing wave model resolution versus a probabilistic forecast system in this specific scenario - where approximately an order magnitude increase in computing power has been available. For such a study, the authors have done a good job with being concise in their use of probabilistic verification metrics and delivered a clear set of results.

However, I would recommend that publication is made subject to a number of major

revisions. These are required in order to address a number of questions raised by the study, but which the authors have only dealt with very briefly or passed over: 1. For a wind driven wave model, the nature and quality of the forcing winds are a key consideration in the model performance. The driving wind model therefore needs to be well documented and, specifically for this paper, any differences in horizontal resolution associated with the deterministic and ensemble forecasts systems need to be provided clearly in Section 2 These were not clear to me on my read through, and I am left with the impression that the authors have compared a 10km wave model with a 5km wave model but using a similarly specified wind model for both deterministic and ensemble forecasts?

\*\*\* The HIGH and LOW deterministic models are forced with wind from 3 km DMI-HIRLAM, while LOWENS are forced with wind from 5 km ensemble DMI-HIRLAM. Note that this disfavours the LOWENS forecasts, since the ensemble runs are forced by lower-resolution wind fields.

We will modify the manuscript to make clearer which wind forcing is used for the different configurations. \*\*\*

2. If this is indeed the case, then I think the wind forcing being used, wave model resolutions chosen and available observations naturally lean the study toward favouring the ensemble. This is acceptable, but needs to be acknowledged and discussed further within the paper. From a wind perspective, if no higher resolution atmosphere model that will improve representation of the land-sea boundary layer is available then the ensemble's provision of multiple answers will generally help the verification scores from that system. Whilst a costly enhancement, the change from (LOW) 10km to (HIGH) 5km resolution may not be enough to significantly enhance wave forecast performance in the coastal zone and, besides, only one observation site is available to illustrate coastal performance. This means that it is difficult for the reader to get a clear picture of what advantages the HIGH res model is expected to yield - I'd suggest that might be improved by some visualization of model fields in order that the impact of changes

from LOW to HIGH over the wider region can at least be illustrated.

\*\*\* We do not understand why the referee thinks that our set favours the ensemble. In our opinion, the setup disfavours the ensemble, as pointed to under 1.

The Land-sea boundary problem has been accounted for using water-wind, as described in line 103.

To illustrate the performance of the different configurations, we will in the revised manuscript show forecast fields during the Toni storm, January 2017. \*\*\*

3. Although, in my view, the experiments favour the ensemble system, the paper still raises a valid point: which is that when using regular grid wave models and an order of magnitude computing resource to invest then the ensemble will likely provide a better return, in terms of improving forecast skill over the larger offshore part of the domain. However, in order to make this point the authors also need to be mindful of and discuss the study within the context of rather more of the open literature than they have done. For example, Cavaleri et al (2018) provide an exhaustive discussion of coastal processes and how wind and wave models need to improve in order to properly represent these - it would be good if the authors can set out where and how the HIGH system attempts to address these aspects of coastal forecasting better than the LOW or LOWENS systems. Similarly, there is also the question of whether an unstructured or refined grid approach would enable significant improvements in coastal regions of the domain whilst keeping the model efficient offshore and enabling a best of both worlds approach (e.g. Bunney and Saulter" 2016). So I would recommend that the authors try to address these aspects of the paper with appropriate references in both Sections 1 and 6.

\*\*\* Thanks for pointing to these aspects. We will include these in the discussion and may be also in the introduction. \*\*\*

SPECIFIC COMMENTS

Paragraphs at line 44 and 48. I think this discussion could be a bit more expansive? The authors have followed through the practical viewpoint where the wave model is scaled to the NWP and then resolution is increased if there is spare resource. This is a quite standard 'in practise' way of working, but as a motivating point it would be good if the authors could expand on what scales they believe are required for an idealised/ pragmatic wave forecasting system that dealt with both coastal and offshore areas of the region.

\*\*\* We do not believe that there is one simple model setup, performing well both for offshore and coastal conditions. One has to resort to nested configurations, unstructured mesh, or subgrid-scale representations. We will expand the discussion to include these aspects. \*\*\*

Sentence at line 64. I'm not convinced that the ensemble vs resolution increase argument is generic, rather it depends on where the model is being used and how end-users will deal with the resulting products. So I think it would be better to contextualise this argument to the situation in question - a wind-wave dominated regional sea with a mixture of offshore and coastal regimes.

\*\*\* We do not quite understand this point. At line 64 we mention two ways to spend additional computer resources: ensembles or increased resolution. We do not bring forward any arguments or analysis at this point. \*\*\*

Sentence at line 112. I'm not convinced the information about the spin-up is that useful.

\*\*\* In that case we will remove it \*\*\*

Paragraph at line 117. Around here would be an excellent place to add further detail regarding the NWP forcing.

\*\*\* We will add info on wind resolution to table 2. \*\*\*

Paragraphs at lines 247 and 255. The dependencies of RMSE/bias on SWH are to be expected when matching up deterministic forecasts since small timing errors in the

predicted wave time-series will have larger impacts on the model-observation match-up in the upper percentiles of the SWH pdf than in the lower percentiles.

*** Yes, we will do a remark on this. ***

Section 6.1. It would be useful to state the resolution of the NWP systems underpinning Tuomi et al.'s wave models.

*** We will do so ***

Section 6. For completeness it would be worth discussing the spread-skill character-istics of the LOWENS system. At the sort of short forecast ranges discussed, these systems are usually under-spread and it would be useful to know if this is also the case here (and if not, why not?). The ability of the ensemble to properly generate spread provides the difference between running a system that provides some improvements to forecast verification vs a deterministic model through a partial sampling of forecast uncertainty, and one that genuinely samples the likely observed outcomes.

*** We will show rank histograms and discuss these. An example of rank histogram is shown in Fig 1. ***

Section 6. This would be a good place to talk through the computational limits placed by using a regular grid scheme in this region and some of the other modelling options that might allow some best of both worlds solution to be achieved in future. Its fair to say that in supercomputing terms a resource increase of order 3-10 times might be the maximum expected over 1 or 2 new systems, so the problem highlighted here is important.

*** We think we have already replied to the issues of nested domains, unstructured grids etc. above. ***

**Fig. 1.** Histogram over the rank of the SWH observation relative to the ensemble members. Darss Sill, 48 hr forecast time.

---

## Author Response (AR1)

Dear editor,

We appreciate very much the comment from the two anonymous referees, which have been extremely useful for improving the manuscript. Below, you find the referees' comments with our responses in *italic*. Section numbers and line numbers refer to the revised manuscript.

**Response to Anonymous Referee #1**

**1 General comments**

The paper addresses a topical issue in the operational oceanography in the marginal seas – whether to introduce ensemble forecasting. The Authors have run the wave model WAM for the Baltic Sea with different horizontal and spectral resolutions and different atmospheric forcing to study whether one should increase resolution or introduce ensembles to provide better forecast accuracy. The question is interesting, but one would expect a more thorough and systematic approach in building and introducing the model and forecast system configurations and in analysing the results.

It has been long known, although not perhaps explicitly said, that the open sea areas of the Baltic Sea, where the shallow water effects can be neglected, do not much benefit of reducing the grid size. That said, there are several areas, where the high resolution is important to solve the shallow water effects and address the effects of islands and irregular shoreline on the wave fields.

> *It may be common knowledge that for open sea areas in the Baltic Sea, wave forecasts will not benefit from (further) decrease of the grid size, '..although perhaps not explicitly said'. It is not clear to us what the intention of this remark is; we find it a perfect motivation for our study. We have not been able to find any reference that states this, thus we have not been able to include this view. If the referee can help us in this matter, we would be happy to include such references.*
> .

Due to the small size of the Baltic Sea, the wave field is dominated by the wind waves and the accuracy of the wave forecast is largely dependent on the accuracy of the atmospheric forcing. Therefore comparing systems run with wind forcing from different NWP systems to address the question about choosing between ensembles and resolution is not entirely valid. Also the earlier studies the Authors refer to in the discussion most likely have different/older versions of the WAM model. Therefore the differences or non-differences cannot directly be connected to resolution or the atmospheric forcing. And also, if the time periods used in verification are relatively short (2-3 years) and different ones, the inter-annual variability in the wind conditions might also affect the accuracy. I'd expect more discussion about these subjects.

*We agree that wind waves dominate the Baltic Sea and therefore comparing wave forecasts driven by wind forcing from different NWP-systems will not be entirely valid. However, due to an unfortunate error in (old) table 1, there has been a misunderstanding about the wind forcings in our study. We will explain this below and have revised the manuscript and in particular table 1 accordingly.*

*Discussion of length of verification period has been included in discussion section. Btw., our three-years verification period is not short compared to other published work. For instance,* (Bunney and Saulter 2015) *use eight months,* (Pezzutto et al. 2016) *use seven months, and* (Cao et al. 2009) *use 12 months.*

The only driving factor in reducing the grid size in the open sea wave modelling is not the accuracy of the wave forecast. There might also be other factors. For example coupling of wave and 3D ocean models might benefit of having high enough resolution. Same applies also for atmosphere – wave coupling. Furthermore, the benefits of higher resolution come also when using high-resolution wind fields nowadays available for the Baltic (e.g. HARMONIE with 2.5 km resolution), which are not possible to get full benefit from if wave model resolution is coarser. I'd also like to see more discussion related to these subjects.

*A discussion of higher resolution required by two-way coupling to atmospheric model and ocean model is certainly relevant. Similarily, we agree that increasing the horizontal resolution of the NWP-system may lead to better wind forecasts, in particular due to better descriptions of processes in extratropical cyclones, but we think this has already been mentioned in lines 52-58.*

Also, I think that the title should include indication, that you are focusing on the open sea, deep water areas.

*We find our title catchy an fully covering our aim of the study, since we not a priori limited ourselves to the open sea areas. This was dictated during the work, due to available observations. Therefore, we have instead modified the conclusion and the abstracts and put more emphasis on the conclusion for the offshore sea.*

Is same wind forcing used both for HIGH and LOW NSB grids? This is not explicitly said in the manuscript. And is the forcing used the deterministic ECMWF or the HIRLAM wind field? Table 1 mentions both HIRLAM and ECMWF and Table 2 only states that one ensemble member is used as forcing, but not indicated whether the 1 ensemble is the ECMWF or HIRLAM deterministic forecast or something else. If HIRLAM is used for the HIGH and LOW NSB grids, then you are comparing wave model results with different NWP forcings against each other. Is it then question about resolution or different wind forcings? I suggest that you run both HIGH and LOW NSB grids using the control forecast from the ECMWF ENS system and compare the difference between them and the LOWENS to find what type of effects the resolution and introducing ensembles causes to the system. Furthermore, it is of course interesting to see, if with higher resolution wind forcing (e.g. HIRLAM) the results would further improve. If and when you use the HIRLAM forcing for the wave models, please specify, how you process the wind fields with 3 km resolution to the wave model grid with 5 km (and/or 10 km) resolution.

*All configurations in the NSB-domain are forced with wind from the DMI-HIRLAM NWP-system. For the HIGH and LOW configurations the horizontal resolution is 3 km, while for the LOWENS it is 5 km. The ECMWF-forced North Atlantic domain is deterministic in all cases, and serves only as boundary data.*

*Unfortunately, there was an error in in table 1 in the oroginal manuscript and this table has been corrected and extended in the revised manuscript.*

*We have also included a sentence on how the 3/5 km HIRLAM wind fields are transformed to 5 and 10 km WAM grids by bilinear interpolation.*

Also, you should separately check, what can be addressed to spectral resolution and what to grid resolution. And also please check other parameters than SWH, for example it would be interesting to see, if there are affects to wave periods or directions, when using higher spectral resolution.

*It could certainly be interesting to check the effect of changing spectral resolution and grid size separately, but would require a lot of additional work in the form of dedicated runs and analysis. The scope of the present work is to use different configurations already running in an operational environment to gain some knowledge regarding the resolution-ensemble issue. For the same reason, we can not isolate the effect from higher spectral resolution on wave periods and – directions.*

When looking through the supplement material, I was bit confused, why Arkona and Vahemadal were chosen to be the stations shown in the manuscript. E.g. looking Fig S2 Finngrundet, Nothern Baltic, Huvudskar show that HIGH gives lower rmse in many cases for the higher (of over 3 m) significant wave heights than LOW or LOWENS. If the lower rmse of LOWENS over longer forecast ranges come mainly from forecasting smaller than 3 m SWH it might not be that useful for duty forecasters. This type of conditions typically do not affect the marine traffic or the offshore structures, it is the extremes. Therefore it would be important to see how the different forecast systems behave in high wave conditions. The time period used in this study contains at least the January 2017 storm. It would be interesting to see a detailed comparison of the results during this storm and also in some other high wind events.

*Vahemadal was chosen because it stands out against all other stations in the analysis (HIGH forecasts performs better for this station), and Arkona because for this station, together with Darss Sill, the ENSMEAN performed better than the other forecast classes, and this result is statistically significant (non-overlapping confidence bands). For the four last stations, the ENSMEAN perform best, but this result is not statistically significant. We have modified parts of (new) Section 6 the manuscript to make this clearer.*

*Remember, however, that for SWH above 3 m there are few observations/forecasts and the statistics becomes very uncertain, as shown by the large error bars on Fig. S3 and S4, and also mentioned in the text.*

*Introducing the 11 January 2017 'Toini' storm as a case is a good suggestion. We have devoted a section to this in the revised manuscript, including and discussing the plot shown in Fig. 9 of SWH during January 2017, 48 hour forecast for Northern Baltic.*

It is good that the Authors have shown that with ECMWF ENS forcing the accuracy of the wave forecasts is ok in the open sea areas of the Baltic Sea. I suggest that the authors do the more comprehensive model runs suggested above and also more detailed analysis of the results and also discuss the advantages and disadvantages of each system more thoroughly. Furthermore, it would be interesting to see how much skill the ensemble forecasts have for longer forecast period. To my experience, there is not much spread in the ensembles for the first two or three days and the true benefits of the ensemble system and probabilistic forecast usually comes with longer forecast ranges. It would be interesting to see up to which forecast lengths the ensemble system shows skill in forecasting the Baltic Sea wave conditions both in average and extreme conditions.

*As explained above, our runs are all forced with DMI-HIRLAM ensembles, and therefore they only reach 48 hour forecast time.*

*The experience that the benefits of an ensemble system shows up after three days only may be true for ECMWF-ENS, but for the DMI-HIRLAM ensemble we see an effect for some stations already from about 36 hours (see Fig. S2)*

Please also see my specific comments given below.

**2 Specific comments**

**2.1 Introduction**

Lines 29-35: I'd expect that the concept of deep and shallow water waves is introduced here, since this is one of the key issues in the discussion of the results.

*Agree, we have done this.*

Line 33: Bathymetry is important only if waves interact with bottom.

*Agree, we have introduced a more precise formulation*

Lines 29-25: How about weak non-linear wave-wave interactions?

*Agree, (Non-linear) wave-wave interaction has been mentioned here*

Lines 41-42: Seasonal ice conditions vary quite a lot in the Baltic. Perhaps this description refers to an average ice winter?

*Yes, this is included to remind the reader that sea ice is an issue in the Baltic Sea.*

Line 51-52: Is Baltic Sea shallow considering the average wave conditions? If then the use of higher resolution should make a difference, which is not in agreement with the conclusion drawn by the Authors later on. Baltic Sea is shallow compared to the Oceans, but when considering the typical wave periods/lengths in the Baltic, in most cases waves in the open sea areas are deep water waves, expect for high and extreme wave conditions.

*The Baltic Sea may be 'shallow' compared to the world ocean, but is in general not shallow as regard wave conditions. We have therefore deleted the misleading sentence.*

**2.2 Model and setup**

This section needs restructuring. All information needed is basically given, but the order of things and the fact that some information is only given in Tables and the table is not referred in the corresponding place in text makes it difficult to follow.

*We have re-structured section 2. incl. the tables to make it more understandable.*

Also please define explicitly, which wind forcing is used for LOW and HIGH configurations. Table 3 indicates that deterministic ECMWF forcing is used for the coarse, larger domain and HIRLAM (and possibly also ECMWF?) for the smaller high resolution domain.It is not clear to me if this Table refers to DMI operational setup or for the setup used in this paper.

*We have corrected errors in Tables 1 and 2 regarding the wind forcing in the different configurations, as explained above.*

Lines 73-77: Please specify the source terms and formulations used in the model runs.

*Source terms are described on lines 105-110, so we do not understand the referee's remark.*

Lines 78-82: Specify the horizontal resolution of the areas already here or cite a Table where they are given. I also suggest adding the resolution info to Table 1.

*Horizontal resolutions are now given in Table 2.*

Line 88: Specify the various sources used to compile the bathymetry

*These are now listed in Table 2.*

Line 121-122: You use only 11 members of the total 50 available from ECMWF. How do you select, which members you use?

*As explained above, we use DMI-HIRLAM atmospheric ensemble forcing, of which a subset of 11 ensemble members are run at routine basis at DMI. The ensembles are generated by perturbing a number of processes, e.g. cloud physics, which do not have a direct impact on the wind field. A subset of 11 ensembles was recommended by the DMI-HIRLAM ensemble developers to cover the spread in surface wind.*

Table 2: It is unclear to me what the column 'Ensemble members' mean for LOW ad HIGH.

*'1' meant deterministic forecast. We have replaced by '-' in Table 2*

**2.3 Observations**

Why not use Helsinki wave buoy data from Gulf of Finland? This should be available through CMEMS. Helsinki site mostly represent deep water conditions and it would be interesting to see, how the setups behave there compared to Vahemadal.

*Helsinki wave buoy does not have many valid data in our verification period, see plot below. Therefore it is not included.*

[Figure]

Table 3 gives only model depth at the buoy locations. It would be important to know also the actual depth at the buoy locations to evaluate, whether the model is adequately able to account for the deep and shallow water features in the wave field.

*The actual water depths have been listed in Table 3.*

It is bit unclear to me, what is the function of Figure 3. The details are lost here, since the images are so small. If they area meant to represent the overall description of the wave conditions at each site, please also (or maybe instead) give some description in the text. And if it is to show the gaps in the measured data, that could be put in a table.

*We have moved Figure 3 to the supplementary information; we think the reader should have the opportunity to see which data series were actually used. To give the reader a feeling of the data, we have in the main text replaced figure 3 by a new table 4 showing frequency of swh-height in different intervals.*

**2.4 Verification**

Give some explanation, why you have selected Bothnian Sea, Arkona and Vahemadal stations for more detailed analysis

*See above*

I also suggest doing some verification of the forcing wind fields.

*We have included wind verification for two Danish coastal stations in a new Section 5.*

In addition to verifying the general accuracy, I'd expect to see some verification of high wind/wave events. They are the most important ones to forecast accurately considering the marine traffic and offshore structures.

> *Sections 5.1.1 (RMSE and BIAS as function of SWH) and 5.3 (Brier score verification) are already verifications of high wave events. In addition we have included an intercomparison of forecasts of the Jan 2017 Toini storm.*

I would also expect more discussion of the importance of wind field accuracy on the accuracy of the wave forecast. The accuracy of wave forecast in the open sea areas might not benefit from higher resolution in the wave model grid, but what about when the wind forcing has high resolution, such as the HARMONIE forecasts run for the Baltic with 2.5 km resolution in several of the MET services. In order to account for the benefits of this, higher resolution in wave model grid might become important.

> *This point is addressed at the end in the Introduction.*

**2.5 Discussion**

You should very carefully analyse and explain, what you are actually comparing in Table 2. To my understanding you are comparing wave forecast system, which have different resolutions, wind forcing and also most likely different wave model versions. So the differences in accuracy cannot solely be attributed to resolution.

> *We think the referee refers to table 4 (present table 6). In that case we agree and have put in an appropriate sentence in the discusion*

Table 4 – Why have you not calculated rms errors for the Helsinki wave buoy for LOW, HIGH and HIGHENS?

> *Due to lack of observations, see above.*

I'm not sure why the ice coverage is discussed here. You are comparing the forecasts against buoy measurements and the buoys are recovered well before there is a risk of ice in the area. Therefore handling of ice should not cause any problems in your verification results. That said, you of course have this element during the season and in the areas where you are unable to do the verification. You could also give a short description of the ice conditions in 2015-2017 so that readers would be able to evaluate, how big effect this might be.

> *The effect of ice cover is described because it is a potential systematic error source of the wave forecasts, and in addition may introduce a bias in the verification due to withdrawal of observing*

*buys during winter time with high waves. We have extended the discussing, ending up with making it likely that 'sea ice problems' are small during the period 2015-2017.*

**Response to Anonymous Referee #2**

OVERVIEW:

This paper will serve a useful purpose in documenting performance of an operational wave forecast modelling system for the Baltic and in assessing and discussing the relative benefits of increasing wave model resolution versus a probabilistic forecast system in this specific scenario - where approximately an order magnitude increase in computing power has been available. For such a study, the authors have done a good job with being concise in their use of probabilistic verification metrics and delivered a clear set of results.

However, I would recommend that publication is made subject to a number of major revisions. These are required in order to address a number of questions raised by the study, but which the authors have only dealt with very briefly or passed over:

1. For a wind driven wave model, the nature and quality of the forcing winds are a key consideration in the model performance. The driving wind model therefore needs to be well documented and, specifically for this paper, any differences in horizontal resolution associated with the deterministic and ensemble forecasts systems need to be provided clearly in Section 2 These were not clear to me on my read through, and I am left with the impression that the authors have compared a 10km wave model with a 5km wave model but using a similarly specified wind model for both deterministic and ensemble forecasts?

> *The HIGH and LOW deterministic models are forced with wind from 3 km DMI-HIRLAM, while LOWENS are forced with wind from 5 km ensemble DMI-HIRLAM. In our opinion, this disfavours the LOWENS forecasts, since the ensemble runs are forced by lower-resolution wind fields. We have modified Section 2 of the manuscript to make clearer which wind forcing is used for the different configurations.*

2. If this is indeed the case, then I think the wind forcing being used, wave model resolutions chosen and available observations naturally lean the study toward favouring the ensemble. This is acceptable, but needs to be acknowledged and discussed further within the paper. From a wind perspective, if no higher resolution atmosphere model that will improve representation of the land-sea boundary layer is available then the ensemble's provision of multiple answers will generally help the verification scores from that system. Whilst a costly enhancement, the change from (LOW) 10km to (HIGH) 5km resolution may not be enough to significantly enhance wave forecast performance in the coastal zone and, besides, only one observation site is available to illustrate coastal performance. This means that it is difficult for the reader to get a clear picture of what advantages the HIGH res model is expected to yield - I'd suggest that might be improved by some visualization of model fields in order that the impact of changes from LOW to HIGH over the wider region can at least be illustrated.

> *We do not understand why the referee thinks that our configurations favours the ensemble. In our opinion, the setup disfavours the ensemble, as pointed to under 1.*

> *The Land-sea boundary problem has been accounted for using water-wind, as described in the manuscript.*

*To illustrate the performance of the different configurations, we have included in Section 2 forecast fields valid at the peak of 'Toini' storm, January 2017.*

3. Although, in my view, the experiments favour the ensemble system, the paper still raises a valid point: which is that when using regular grid wave models and an order of magnitude computing resource to invest then the ensemble will likely provide a better return, in terms of improving forecast skill over the larger offshore part of the domain. However, in order to make this point the authors also need to be mindful of and discuss the study within the context of rather more of the open literature than they have done. For example, Cavaleri et al (2018) provide an exhaustive discussion of coastal processes and how wind and wave models need to improve in order to properly represent these - it would be good if the authors can set out where and how the HIGH system attempts to address these aspects of coastal forecasting better than the LOW or LOWENS systems. Similarly, there is also the question of whether an unstructured or refined grid approach would enable significant improvements in coastal regions of the domain whilst keeping the model efficient offshore and enabling a best of both worlds approach (e.g. Bunney and Saulter„ 2016). So I would recommend that the authors try to address these aspects of the paper with appropriate references in both Sections 1 and 6.

*Thanks for pointing to these aspects. We have included a short summary of these aspects in the conclusion section.*

SPECIFIC COMMENTS

Paragraphs at line 44 and 48. I think this discussion could be a bit more expansive? The authors have followed through the practical viewpoint where the wave model is scaled to the NWP and then resolution is increased if there is spare resource. This is a quite standard 'in practise' way of working, but as a motivating point it would be good if the authors could expand on what scales they believe are required for an idealised/ pragmatic wave forecasting system that dealt with both coastal and offshore areas of the region.

*We do not believe that there is one simple model setup, performing well both for offshore and coastal conditions. One has to resort to nested configurations, unstructured mesh, or subgrid-scale representations. Instead of in the introduction, we have incorporated these aspects in the conclusion section (see above).*

Sentence at line 64. I'm not convinced that the ensemble vs resolution increase argument is generic, rather it depends on where the model is being used and how end-users will deal with the resulting products. So I think it would be better to contextualise this argument to the situation in question - a wind-wave dominated regional sea with a mixture of offshore and coastal regimes.

*We do not quite understand this point. At line 64 we mention two ways to spend additional computer resources: ensembles or increased resolution. We do not bring forward any arguments or analysis at this point.*

Sentence at line 112. I'm not convinced the information about the spin-up is that useful.

*OK, we have removed it.*

Paragraph at line 117. Around here would be an excellent place to add further detail regarding the NWP forcing.

*We have added info on wind resolution to table 2.*

Paragraphs at lines 247 and 255. The dependencies of RMSE/bias on SWH are to be expected when matching up deterministic forecasts since small timing errors in the predicted wave time-series will have larger impacts on the model-observation match-up in the upper percentiles of the SWH pdf than in the lower percentiles.

*Yes, this is now mentioned in the conclusion*

Section 6.1. It would be useful to state the resolution of the NWP systems underpinning Tuomi et al.'s wave models.

*We have added that info to (present) Table 6.*

Section 6. For completeness it would be worth discussing the spread-skill characteristics of the LOWENS system. At the sort of short forecast ranges discussed, these systems are usually under-spread and it would be useful to know if this is also the case here (and if not, why not?). The ability of the ensemble to properly generate spread provides the difference between running a system that provides some improvements to forecast verification vs a deterministic model through a partial sampling of forecast uncertainty, and one that genuinely samples the likely observed outcomes.

*We have included rank histograms as Section 6.4 and discuss these.*

Section 6. This would be a good place to talk through the computational limits placed by using a regular grid scheme in this region and some of the other modelling options that might allow some best of both worlds solution to be achieved in future. Its fair to say that in supercomputing terms a resource increase of order 3-10 times might be the maximum expected over 1 or 2 new systems, so the problem highlighted here is important.

*We think we have already replied to the issues of nested domains, unstructured grids etc. above.*

[revised manuscript text omitted]

**Figure S 12 The average ice cover for February: a) 2015, b) 2016, c) 2017 and d) average 2010-2018.**

---

## Referee Report (RR1)

**Better Baltic Sea Wave forecasts: Improving resolution or introducing ensembles?**

**Reviewed at iteration:** Revised Submission

**Recommendation:** paper is accepted subject to minor/technical revisions.

**Generic remarks:** Thanks a lot for the authors for taking on board the previous review comments. In particular the improved documentation of the atmospheric forcing used by the wave models has made the paper and its conclusions a lot clearer. I would be happy to see this published without a further review subject to some technical (written English) corrections suggested below. However, I'd also like the authors to consider some potential minor revisions to the manuscript.

**Potential minor revisions:**

Section 2: Please confirm in the text whether any of the source term tuning parameters are different or the same in the various wave configurations. Previous review comments had also asked for the WAM version(s) to be documented.

Section 2: Is it possible to add some justification as to why the authors might consider the 3km atmospheric system to offer a step change in wind forcing skill over the 5km configuration that underpins the ensemble – this is not an enormous resolution change, so which processes are believed to be improved? Ideally provide references for the atmosphere systems in question.

Section 5: Not suggesting that additional results/tables are presented – but is it possible to comment on the performance of the S05 atmosphere model's control member performance? Is it similar to, better or worse than S03? This would help put the effect of the ensemble into context.

Section 6: As per section 5, if it is possible to comment on performance of S05's control member relative to the LOW wave model that will help contextualise the ensemble.

Figure 9: Is it possible to show the range of LOWENSMEAN forecasts – it would be good to see if any of the ensemble members predicted the peak in SWH.

Section 7 (or 8): I'd still be keen to see a little more discussion on the systems, what has been verified and what that means in terms of application. Now that the source of wind forcing has been stated more explicitly, what we have are: LOW – a system with 3km winds integrated onto a 10km wave grid; HIGH – 3km winds integrated onto a 5km grid; and LOWENS – a system with 5km winds integrated on a 10km grid but run as an ensemble in order to sample uncertainty in the atmosphere model and evolution of the meteorological conditions. Validation is carried out on a site specific basis. So, the important points to consider are:

1. Whilst the validation is a very standard approach for wave models, it is actually documenting site-specific forecast performance for the model. This is an important consideration when testing high resolution systems, where 'double-counting errors' start to get introduced and site specific verification does not improve as a result. On the other hand, the majority of forecast use for these models are site-specific, so the results are absolutely valid when one considers the products that get generated from these models.

2. The comparison between LOW and HIGH suggest that integrating the 3km winds onto a coarser wave model grid has no impact in terms of the verification at offshore sites. Why might this be? Speculating, this is probably because wave development is a function of winds, but integrated over a longer fetch area – so the resolution of the atmosphere model is (within reason) perhaps less important than the model's ability to properly place major synoptic features. Figure 2 hints at this, since there appears to be little addition structure in the HIGH model field offshore compared to LOW.

3. The ensemble mean gives the best site-specific forecasts offshore, in terms of limiting the overall error. This is despite using a coarser resolved atmospheric model and the coarser wave model. This is consistent with the argument above that uncertainty large scale feature development is perhaps (generally) more important to wave forecasts in open waters than getting the momentum exchange associated with small scale atmospheric features correct. I'd suggest that argument is further supported if the LOWENS control member verification at the offshore locations is not significantly worse than for LOW and HIGH.

I appreciate it if points 2 and 3 feel too speculative for the authors to consider including in their discussion, but I would advocate some comment along the lines of point 1.

Section 8: The under-spread in the ensemble suggests that there is scope for improving that system. I'd suggest this is a valid conclusion.

**Technical revisions:**

Line 31-32: dissipation of the wave energy mainly occurs through internal processes, e.g. whitecapping.

Line 48-49: The equations of the NWP model are discretized on a horizontal grid with a certain spatial resolution, which influences the maximum spatial resolution of the wave model.

Line 52: Over time, technical development has increased available computational resources, making it possible to increase…

Line 75: …modelled sea-surface temperatures (SSTs) by the NEMO…

Line 76-77: Introducing such coupling may demand a high horizontal resolution, in atmosphere, wave and ocean models, in order to describe the fluxes most satisfactorily.

Line 84: …wind forecasts is in Section 5, whilst verification of a principle wave forecast variable, significant wave height (SWH), is presented in Section 6.

Line 123: Each forecast run…

Line 133: …with characteristics identical to LOW, but using a parallel…

Line 167: …the area with SWH above 6m extends further southward…

Line 170-171: Observed series of SWH from wave measurement sites in the Baltic Sea, obtained from the Copernicus Marine Environment Monitoring Service (CMEMS) database, are used.

Line 306: …HIGH forecast has a significantly smaller under-prediction bias than the other forecast classes.

Line 364: The conclusions hold,…

Line 410: …field approaches an ice-covered area,…

Lines 412-413: …when dense enough, acts as a solid shield that effectively removes all local wave energy…

Line 414: …thick enough for this to be approximately correct.

Line 445-446: …there are no indications that a further increase of the WAM model will result in enhanced site-specific forecast performance.

---

## Author Response (AR2)

Dear editor,

Below, you find the referees' comments to the first revision of our manuscript with our responses in *italic*.

**Response to Referee #1**

The manuscript has improved a lot. I have some minor comments that I suggest to be taken account before publishing the manuscript.

The language of the manuscript should be checked by a native English speaker. There were also some typos.

*Our english-born collegue corrected the english language in the manuscript.*

Line 26: I suggest changing "severe surface waves" to "severe wave conditions"

*OK - done*

Lines 72-81: This should be in Discussion-section

*OK - done*

Lines 123-125: Is the North Atlantic grid forced by ECMWF-HRES also run 4 times a day? To my understanding the forcing is available only twice a day.

*Yes, it is run four times/day, even if a new forcing is available only twice/day.This has been specified in the text.*

Lines 153-154: If you would only increase grid resolution, not the spectral resolution. How would the computational efforts then compare. If you can not add an exact number, maybe some discussion about this would be appropriate. This experiment setup does not show, whether the possible benefits of the higher resolution comes from increasing horizontal or spectral resolution.

*We have split the timing calculations into two parts in the manuscript, one for the spatial resolution, and one for the spectral resolution.*

Lines 174-175: I again comment the selection of buoys used for comparison. Your requirement of more than 40% of temporal coverage basically leaves out all the buoys that are in the areas, where the seasonal ice cover typically ranges from Dec/Jan to May, such as the Bothnian Bay and Gulf of Finland.

Maybe you should use this criteria in a way that accounts for the time at each buoy location the sea is ice-free.

[Figure]

*We have considered the two suggested stations, Bothnian Bay and Helsinki Buoy/Gulf of Finland, see panels above. As for Bothnian Bay it has data for last half of 2017 only and that's the reason why we still exclude it from the study.*

*Helsinki Buoy has missing data; not only during the ice period (January-May), as expected, but in 2015 until July and with an additional break in October/November, and in 2016 until September. Only in 2017 operation is resumed in May. This not-so-good data coverage during summer may introduce biases in the verification measures, and therefore we continue to exclude station from the analysis. We have made a remark about this in the manuscript.*

*A closer inspection reveals that the reason for the not-so-good data coverage for Helsinki Buoy is corrupt data (constant value) during a part of the period, see panels below.*

[Figure]

Line 176: Comment to the 'sites did not observe the full hour". WaveRiders, which most of the buoys used in validation were, measure waves for 15-30 min period, and calculate the wave parameters based to that measurements. Did you check, whether the timestamp in the measured data files was the starting time, mid-time, or end time of this measurement period?

*No, we did not go into details of the procedures of wave measurements. These may differ among instruments. We assume that the time given is representative for the measurement.*

Table 4: What about higher values than 5m?

*We have added a column '> 5 m' to the table.*

Table 5: Why only Danish stations? I'm not sure that they are the best ones, at least as only ones, to describe the wind field accuracy in the Baltic. At least SMHI and FMI have coastal weather station data available in they open data portals. You should consider using them.

*We have added Swedish and Finish coastal stations and now verify based on eight stations in total. The section on verification has been re-written, and the table replaced by a figure showing total rmse.*

Line 310: According to Björkqvist et aͺlthe NB wave buoy measured 8.0 m , not 'almost 8m' during the storm

*OK -corrected*

Line 362: I would not state that the performance of LOWENSMEAN and LOWENS were "superior" to HIGH. For sure, it was better in many cases.

*OK – we have changed the formulation*

Lines 376-386: I would start with informing the readers that these numbers are not directly comparable. Maybe you could start with lines 384-386.

*OK – we have re-arranged the text.*

Line 398: I would not call Tuomi et al. 2017 a study, rather a standard product validation procedure performed in CMEMS.

*OK – we have changed the text*

**Response to Referee #2**

**Better Baltic Sea Wave forecasts: Improving resolution or introducing ensembles?**

**Reviewed at iteration:** Revised Submission

**Recommendation:** paper is accepted subject to minor/technical revisions.

**Generic remarks:** Thanks a lot for the authors for taking on board the previous review comments. In particular the improved documentation of the atmospheric forcing used by the wave models has made the paper and its conclusions a lot clearer. I would be happy to see this published without a further review subject to some technical (written English) corrections suggested below. However, I'd also like the authors to consider some potential minor revisions to the manuscript.

**Potential minor revisions:**
Section 2: Please confirm in the text whether any of the source term tuning parameters are different or the same in the various wave configurations. Previous review comments had also asked for the WAM version(s) to be documented.

*We made a more precise formulation in the manuscript.*

Section 2: Is it possible to add some justification as to why the authors might consider the 3km atmospheric system to offer a step change in wind forcing skill over the 5km configuration that underpins the ensemble – this is not an enormous resolution change, so which processes are believed to be improved? Ideally provide references for the atmosphere systems in question.

*The difference in horizontal resolution between deterministic and ensemble forecasts are due to heuristics more than to science. The deterministic forecasts were introduces first and 3 km was what was affordable on the computer. Later, the ensemble forecasts were introduces, and here 5 km was regarded as the affordable resolution.*

Section 5: Not suggesting that additional results/tables are presented – but is it possible to comment on the performance of the S05 atmosphere model's control member performance? Is it similar to, better or worse than S03? This would help put the effect of the ensemble into context.

*The main idea of introducing the ensemble approach is to get an overall better forecast based on all ensemble members. The control member does not have a special role.*

Section 6: As per section 5, if it is possible to comment on performance of S05's control member relative to the LOW wave model that will help contextualise the ensemble.

*See above*

Figure 9: Is it possible to show the range of LOWENSMEAN forecasts – it would be good to see if any of the ensemble members predicted the peak in SWH.

*We have added all ensemble members to the figure and added an appropriate sentence in the text.*

Section 7 (or 8): I'd still be keen to see a little more discussion on the systems, what has been verified and what that means in terms of application. Now that the source of wind forcing has been stated more explicitly, what we have are: LOW – a system with 3km winds integrated onto a 10km wave grid; HIGH – 3km winds integrated onto a 5km grid; and LOWENS – a system with 5km winds integrated on a 10km grid but run as an ensemble in order to sample uncertainty in the atmosphere model and evolution of the meteorological conditions. Validation is carried out on a site specific basis. So, the important points to consider are:

1. Whilst the validation is a very standard approach for wave models, it is actually documenting site-specific forecast performance for the model. This is an important consideration when testing high resolution systems, where 'double-counting errors' start to get introduced and site specific verification does not improve as a result. On the other hand, the majority of forecast use for these models are site-specific, so the results are absolutely valid when one considers the products that get generated from these models.

2. The comparison between LOW and HIGH suggest that integrating the 3km winds onto a coarser wave model grid has no impact in terms of the verification at offshore sites. Why might this be? Speculating, this is probably because wave development is a function of winds, but integrated over a longer fetch area – so the resolution of the atmosphere model is (within reason) perhaps less important than the model's ability to properly place major synoptic features. Figure 2 hints at this, since there appears to be little addition structure in the HIGH model field offshore compared to LOW.

3. The ensemble mean gives the best site-specific forecasts offshore, in terms of limiting the overall error. This is despite using a coarser resolved atmospheric model and the coarser wave model. This is consistent with the argument above that uncertainty large scale feature development is perhaps (generally) more important to wave forecasts in open waters than getting the momentum exchange associated with small scale atmospheric features correct. I'd suggest that argument is further supported if the LOWENS control member verification at the offshore locations is not significantly worse than for LOW and HIGH.

I appreciate it if points 2 and 3 feel too speculative for the authors to consider including in their discussion, but I would advocate some comment along the lines of point 1.

*We have added a formulation addressing your point 1 in the beginning of the discussion section.*

Section 8: The under-spread in the ensemble suggests that there is scope for improving that system. I'd suggest this is a valid conclusion.

*We agree and have added a sentence at the end of the conclusion section.*

**Technical revisions:**
Line 31-32: dissipation of the wave energy mainly occurs through internal processes, e.g. whitecapping.

*OK, done.*

Line 48-49: The equations of the NWP model are discretized on a horizontal grid with a certain spatial resolution, which influences the maximum spatial resolution of the wave model.

*OK - done*

Line 52: Over time, technical development has increased available computational resources, making it possible to increase…

*OK - done*

Line 75: …modelled sea-surface temperatures (SSTs) by the NEMO…

*OK - done*

Line 76-77: Introducing such coupling may demand a high horizontal resolution, in atmosphere, wave and ocean models, in order to describe the fluxes most satisfactorily.

*OK - done*

Line 84: …wind forecasts is in Section 5, whilst verification of a principle wave forecast variable, significant wave height (SWH), is presented in Section 6.

*OK - done*

Line 123: Each forecast run…
*Thank you – done*

Line 133: …with characteristics identical to LOW, but using a parallel…

*OK - done*

Line 167: …the area with SWH above 6m extends further southward…

*Thank you - done*

Line 170-171: Observed series of SWH from wave measurement sites in the Baltic Sea, obtained from the Copernicus Marine Environment Monitoring Service (CMEMS) database, are used.

*OK - done*

Line 306: …HIGH forecast has a significantly smaller under-prediction bias than the other forecast classes.

*OK - done*

Line 364: The conclusions hold,…

*OK - done*

Line 410: …field approaches an ice-covered area,…

*OK - done*

Lines 412-413: …when dense enough, acts as a solid shield that effectively removes all local wave energy…

*Thank you - done*

Line 414: …thick enough for this to be approximately correct.

*Thank you - done*

[revised manuscript text omitted]